# Cancer cell type-specific derepression of transposable elements by inhibition of chromatin modifier enzymes
Divyesh Patel [1,2,6], Ville Tiusanen [1], Konsta Karttunen [1,2], Päivi Pihlajamaa [1,3] & Biswajyoti Sahu [1,2,4,5] ✉

Derepression of transposable elements (TE) by epigenetic therapy leads to the activation of immune response in cancer cells. However, the molecular mechanism of TE regulation by distinct chromatin modifier enzymes (CME) in context of p53 is still elusive. Here, we used FDA-approved epigenetic drugs to systematically inhibit distinct CMEs in p53 wild-type and p53-mutant colorectal, esophageal, and prostate cancer cells. We show that distinct TE subfamilies are derepressed by inhibition of different CMEs in cell type-specific manner. Co-inhibition of DNMT and HDAC (DNMTi-HDACi) had the most consistent effect across cancer types. Loss of p53 results in stronger TE activation and TE-chimeric transcript expression and this effect is largely mediated by the non-genomic actions of p53. Robust immune response elicited by DNMTi-HDACi is due to induced inverted repeat Alu expression concomitant with reduced ADAR1-mediated Alu RNA editing. Collectively, our systematic analyses provide insights for rational use of epigenetic therapies in distinct cancers.

Transposable elements (TE) represent more than half of the human genome[1,2] forming a reservoir of gene regulatory elements[3,4]. TEs can act as oncogenic enhancers exploited by tissue-specific transcription factors[5] (TF) and thus TE expression and transposition can be pathogenic[4,6]. In normal cells, TEs are epigenetically repressed by several mechanisms that include DNA methylation[4,7], enrichment of repressive histone marks[4,8], recruitment of Krüppel-associated box zinc finger (KRAB-ZNF)/KRAB-associated protein 1 (KAP1/TRIM28) complex that recruits repressive chromatin modifier enzymes (CME)[9,10], and binding of p53[11].

The combinatorial epigenetic and immune therapy has emerged as a major approach for cancer treatment to overcome the limitations of immunotherapies, evident from improved patient responses[12–14]. SETDB1 amplification is associated with resistance to immune checkpoint blockade therapy, and loss of SETDB1 derepresses TEs encoding viral proteins that activate an immune response[15]. Epigenetic therapy such as inhibition of DNA methyltransferases (DNMTi) or co-inhibition of DNMT and histone deacetylases (DNMTi-HDACi) can activate cryptic promoters within TEs that can splice into nearby protein coding genes, resulting in immunogenic TE-chimeric transcripts that can be targeted by immune therapy[16–18]. Moreover, DNMTi activates TEs that can form double-stranded RNAs

(dsRNAs), mounting a type I/III interferon response inducing viral mimicry that can make cancer cells sensitive to immune therapy[19–21]. However, the molecular mechanisms of TE derepression by distinct epigenetic therapies either alone or specifically in distinct combinations, as well as their cancer cell type-specificity have not been systematically investigated. Importantly, the effect of cellular p53 status on the function of epigenetic drugs is still elusive but critical to investigate, due to the high prevalence of p53 mutations in human cancers. Thus, it is pertinent to delineate the role of p53 in TE regulation in response to epigenetic therapies, given the strong potential of epigenetic inhibitors in cancer therapy.

In this study, we have performed a systematic comparative analysis of TE derepression by four distinct CME inhibitory treatments in five cell lines with different p53 functionalities representing three different cancer types. We show that distinct TE subfamilies are derepressed by CME inhibition (CMEi) in a cell type-specific manner in cancer cell lines. Our analyses revealed that complete loss of p53 or mutant p53 in cancer cells results in stronger derepression of TEs after CMEi. This is intriguingly mediated by the non-genomic actions of p53 in contrast to prior studies looking at p53 response elements in limited set of TEs[22–24]. SETDB1 inhibition (SETDB1i) had a stronger effect on TE expression compared to DNMTi and HDACi

[1]Applied Tumor Genomics Program, Faculty of Medicine, University of Helsinki, Helsinki, Finland. [2]iCAN Digital Precision Cancer Medicine Flagship, University of Helsinki, Helsinki, Finland. [3]Medicum, Faculty of Medicine, University of Helsinki, Helsinki, Finland. [4]Norwegian Centre for Molecular Biosciences and Medicine (NCMBM), University of Oslo, Oslo, Norway. [5]Institute for Cancer Research, Department of Medical Genetics, Oslo University Hospital, Oslo, Norway. [6]Present address: Norwegian Centre for Molecular Biosciences and Medicine (NCMBM), University of Oslo, Oslo, Norway. ✉e-mail: biswajyoti.sahu@ncmbm.uio.no; biswajyoti.sahu@helsinki.fi

treatments alone, and DNMTi-HDACi resulted in the most robust and consistent TE derepression pattern with a synergistic effect. Thus, we focused particularly on the effects of SETDB1i and DNMTi-HDACi on TE activity. A systematic comparison of TE-chimeric transcript expression in different CMEi-treated cells showed that SETDB1i induces expression of TE-derived chimeric transcripts in a cell type-specific manner in cancer cell lines similar to the effect by DNMTi-HDACi seen here in multiple cancer cell lines and as reported earlier in lung, colon and chronic myeloid leukemia (CML) cancer cells[17,18]. This induction is stronger in cells that lack functional p53. In addition, our results show that SETDB1i and DNMTi-HDACi activate an immune response with distinct mechanisms: DNMTi-HDACi results in increased levels of immunogenic dsRNA through the down-regulation of ADAR1 enzyme and decreased Alu RNA editing, whereas SETDB1i upregulates ADAR1 leading to increased Alu RNA editing. In summary, we describe epigenetic mechanisms for CMEi-induced TE derepression contributing to the immune response activation.

## Results

### Inhibition of CMEs results in derepression of common and cell type-specific TE subfamilies

To study the epigenetic regulation of TEs by CMEs in a cancer cell type-specific manner and in the context of p53, we utilized three cancer cell lines derived from colon, esophageal and prostate, where either high somatic retro-transposition events as in esophageal and colon cancers[25] or TEs co-opted as oncogenic enhancers have been reported for example in prostate and colon cancers[5,26]. Specifically, five cell lines with different p53 statuses were used: GP5d colon adenocarcinoma cells (henceforth GP5d) expressing wild-type (WT) p53, GP5d cells with p53-depletion (p53-KO)[27], OE19 esophageal cancer cells harboring a mutation in the exon 9 of *TP53* gene (c.928_930insA, p.N310fs26X)[28], LNCaP-1F5 prostate adenocarcinoma cells, and LNCaP-1F5 cells with p53 deletion. Of note, the *TP53* mutation in OE19 cells results in a truncated p53 protein without a tetramerization domain[29].

For a comprehensive understanding of TE regulation by epigenetic therapies in cancer cells, we utilized FDA-approved compounds targeting the major CMEs involved in repressive epigenetic functions. Specifically, we used Decitabine (DAC) for DNMTi, SB939 for HDACi, and Mitramycin A for SETDB1i to modulate DNA methylation, histone acetylation, and histone methylation, respectively. Of note, DAC was selected over 5-Aza-Cytosine due to its more specific DNA methylation inhibitory activity and reduced toxicity[30] and Mitramycin A over other Mitramycin analogs due to its specificity as SETDB1 inhibitor[31]. Each cell line was treated with individual compounds alone as well as with a combination of DAC and SB939 for DNMTi-HDACi (Fig. 1a; see "Methods" for details) using a 500 nM concentration that has been used for each of these compounds in the previous literature[5,17,18,31,32]. GP5d and OE19 cells were also treated with combination of DAC and Mitramycin A for co-inhibition of DNMT and SETDB1 (DNMTi-SETDB1i) and combination of SB939 and Mitramycin A for co-inhibition of HDAC and SETDB1 (HDACi-SETDB1i). The treatments did not induce considerable cytotoxicity at the concentration used in this study (Supplementary Fig. 1) and their on-target effects were confirmed by (i) reduced DNMT1 level upon DAC treatment, (ii) decreased SETDB1 levels upon Mitramycin A treatment, and (iii) increased H3ac (Pan-acetyl) levels upon SB939 treatment (Supplementary Fig. 2a, b). Following the CMEi treatments, expression of TEs was measured by RNA-seq, and ChIP-seq for active histone marks was used to delineate the epigenetic changes associated with the derepressed TEs (Fig. 1a). RNA-seq data was analyzed for TE expression at the subfamily and individual locus level (Fig. 1a).

Comparative analysis between vehicle-treated (DMSO) GP5d and OE19 cells revealed cell type-specific expression of distinct TE subfamilies with the majority of the differentially expressed TEs from the long terminal repeat (LTR) class (Fig. 1b), in agreement with previous reports from us and others showing a differential TE expression pattern between cancer types[5,26]. The observed differences partly reflect the different p53 statuses of these cell lines, since several LTRs, such as LTR10B and MER61 elements, are known to be enriched for p53 binding sites[33]. These elements showed higher

expression in GP5d cells compared to OE19 with mutated p53, except for MER61E that was upregulated in OE19 cells (Fig. 1b).

Analysis of differentially expressed TE subfamilies induced by CMEi in the five cell lines revealed that the overall effect of DNMTi was weaker compared to HDACi and SETDB1i, with the latter showing the strongest effect (Supplementary Fig. 2b). DNMTi-HDACi synergistically increased the expression of TEs in all cell lines compared to DNMTi or HDACi alone, resulting in 1.5- to 3.2-fold higher number of upregulated TE subfamilies compared to the sum of subfamilies induced by DNMTi and HDACi individually (Supplementary Fig. 2b, Supplementary Data 1). The expression was also modulated by p53: a higher number of TE subfamilies were differentially expressed in p53-KO GP5d and LNCaP-1F5 and p53-mutant OE19 cells compared to WT GP5d or LNCaP-1F5 cells (Supplementary Fig. 2b, Supplementary Data 1).

A total of 128 TE subfamilies were differentially expressed by at least one CMEi treatment in at least one cell line, 111 of which were LTRs (Fig. 1c, Supplementary Fig. 3a, and Supplementary Data 2). Hierarchical clustering of the differentially expressed TE subfamilies revealed two distinct patterns of TE derepression: (i) TE subfamilies that are similarly derepressed in all five cell lines and (ii) TE subfamilies with a cell type-specific response to CMEi (Fig. 1c, and Supplementary Fig. 3a). Particularly, DNMTi-HDACi-treated samples from the five cell lines clustered together, whereas the individual CMEi treatments resulted in a more variable expression pattern in the five cell lines. For example, the majority of LTR12 subfamilies and their associated HERV9 proviruses were derepressed in all five cell lines by DNMTi-HDACi (Supplementary Fig. 3a), but we also detected cell type-specific derepression of distinct subfamilies in cancer cell lines. For example, LTR7Y elements were derepressed by DNMTi in GP5d cells, while DNMTi, HDACi, and DNMTi-HDACi derepressed LTR7Y in OE19 (Fig. 1d). LTR12C and HERV9NC-int showed stronger depression by DNMTi-HDACi than HDACi alone both in GP5d and OE19 cells (Fig. 1d). Collectively, our results show that TE subfamilies are under distinct epigenetic regulation, and that inhibition of distinct CMEs results in derepression of TE subfamilies both in a common and cell type-specific manner in cancer cell lines.

### DNMT and HDAC co-inhibition synergistically derepresses individual TE loci

The effect of CMEi on differential TE expression at individual locus level showed similar changes as observed at the TE subfamily level, with more differentially expressed TEs observed in OE19 cells compared to GP5d cells (Fig. 2a, and Supplementary Fig. 2b). Moreover, HDACi and SETDB1i resulted in stronger derepression of TE loci compared to DNMTi, with at least 12 times greater number of differentially expressed TE loci in GP5d, OE19, and LNCaP-1F5 cells (Fig. 2a). In agreement with earlier reports[18], DNMTi-HDACi derepressed a larger number of TEs as compared to DNMTi or HDACi in the three cell lines. We also compared the regulation of TEs by novel treatment combinations by co-inhibition of DNMT and SETDB1, and HDAC and SETDB1. Both the DNMTi–SETDB1i and HDACi–SETDB1i led to an increase in TE expression in GP5d and OE19 cells, with GP5d interestingly showing a larger number of derepressed TEs with the SETDB1i combinations compared to DNMTi-HDACi (Fig. 2a, and Supplementary Data 3).

Comparison of the major classes of TEs (DNA, LINE, LTR, and SINE) among the differentially expressed TE loci revealed the cell type-specificity of derepression and repression in cancer cell lines (Fig. 2b, and Supplementary Fig. 4a). In GP5d, DNMTi as well as DNMTi-HDACi-induced LTR derepression (40% and 41% of all derepressed elements, respectively), whereas only 19% of the SETDB1i-induced elements were LTRs and 42% SINEs (Fig. 2b, and Supplementary Data 3). Overlap analysis of derepressed TE loci in GP5d, OE19, and LNCaP-1F5 cells showed little overlap between the different CMEi treatments (Fig. 2c, d), suggesting that each CME contributes to the control of a distinct set of TEs. Overall, different CMEs elicit distinct TE expression patterns and DNMTi-HDACi synergistically and robustly derepresses individual TE loci in a cancer cell line-specific manner.

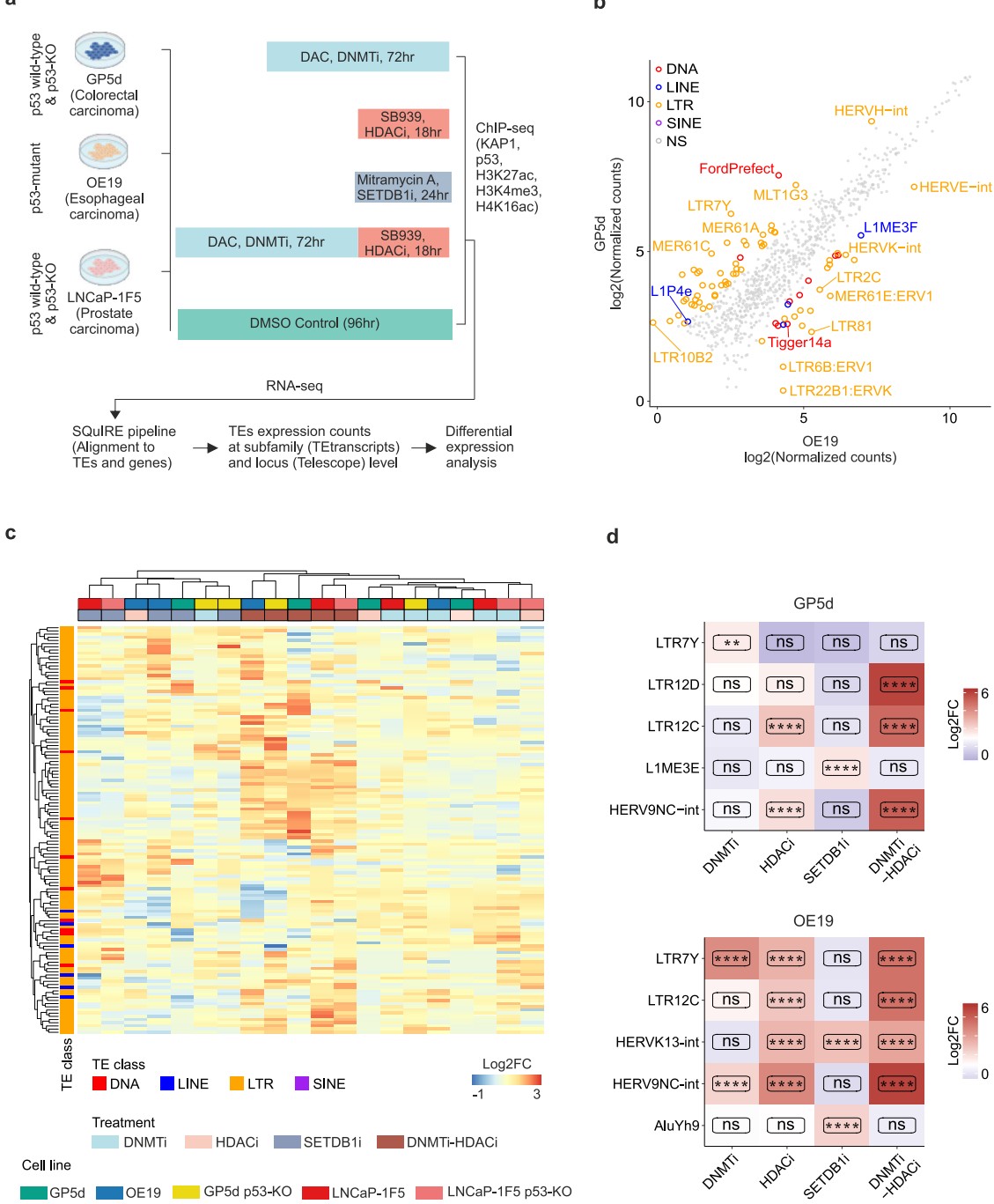

**Fig. 1 | Common and cell type-specific TE subfamilies derepressed by inhibition of CMEs. a** Schematic representation of the CME inhibitor treatments and the analysis pipeline. All five cell lines were treated with DNMTi, HDACi, SETDB1i and DNMTi-HDACi. Additionally, GP5d and OE19 cells were treated with DNMTi-SETDB1i and HDACi-SETDB1i. Decitabine (DAC) DNMT inhibitor, SB939 HDAC inhibitor, and Mitramycin A SETDB1 inhibitor. **b** Differentially expressed TE subfamilies between DMSO-treated GP5d and OE19 cells. Scatter plot shows normalized RNA-seq read counts for TE subfamilies. Differentially expressed TE subfamilies are labeled by TE class. **c** Comparison of differentially expressed TE subfamilies induced by CMEi in the five cell lines. Differential expression analysis

was performed by DESeq2. The heatmap shows log2 fold change (FC) for TE subfamilies with absolute log2FC > 2.5 (treatment vs. vehicle-treated cells from the same cell line) and adjusted p < 0.05 in at least one CME treatment in at least one cell lines. Rows and columns are clustered with hierarchical clustering. **d** Distinct TE subfamilies derepressed by CMEi in GP5d and OE19 cells. Expression changes for TE subfamilies (log2FC) were compared between different CME treatments in GP5d and OE19 cells. Significance symbols: **** indicates p < 0.0001, ***p < 0.001, **p < 0.01, *p < 0.05, ns = non-significant |log2FC| < 1.5 or p > 0.05. Source data are provided as Supplementary Data 11.

## p53 loss results in stronger derepression of TEs upon DNMT and HDAC co-inhibition

Due to the known role of p53 in transcriptional repression of TEs[11], we set out to study the role of p53 in cell type-specific TE regulation upon CMEi. For this, we performed the CMEi treatments in p53-depleted (p53-KO) GP5d and LNCaP-1F5 cells (Supplementary Fig. 5a–c) and compared the TE expression patterns with GP5d and LNCaP-1F5 cells harboring WT p53. Two independent p53-KO GP5d clones showed a strong correlation (Pearson's r = 0.79) in DMSO vs. DNMTi-HDACi expression fold changes, ruling out clone-specific effects (Supplementary Fig. 5d).

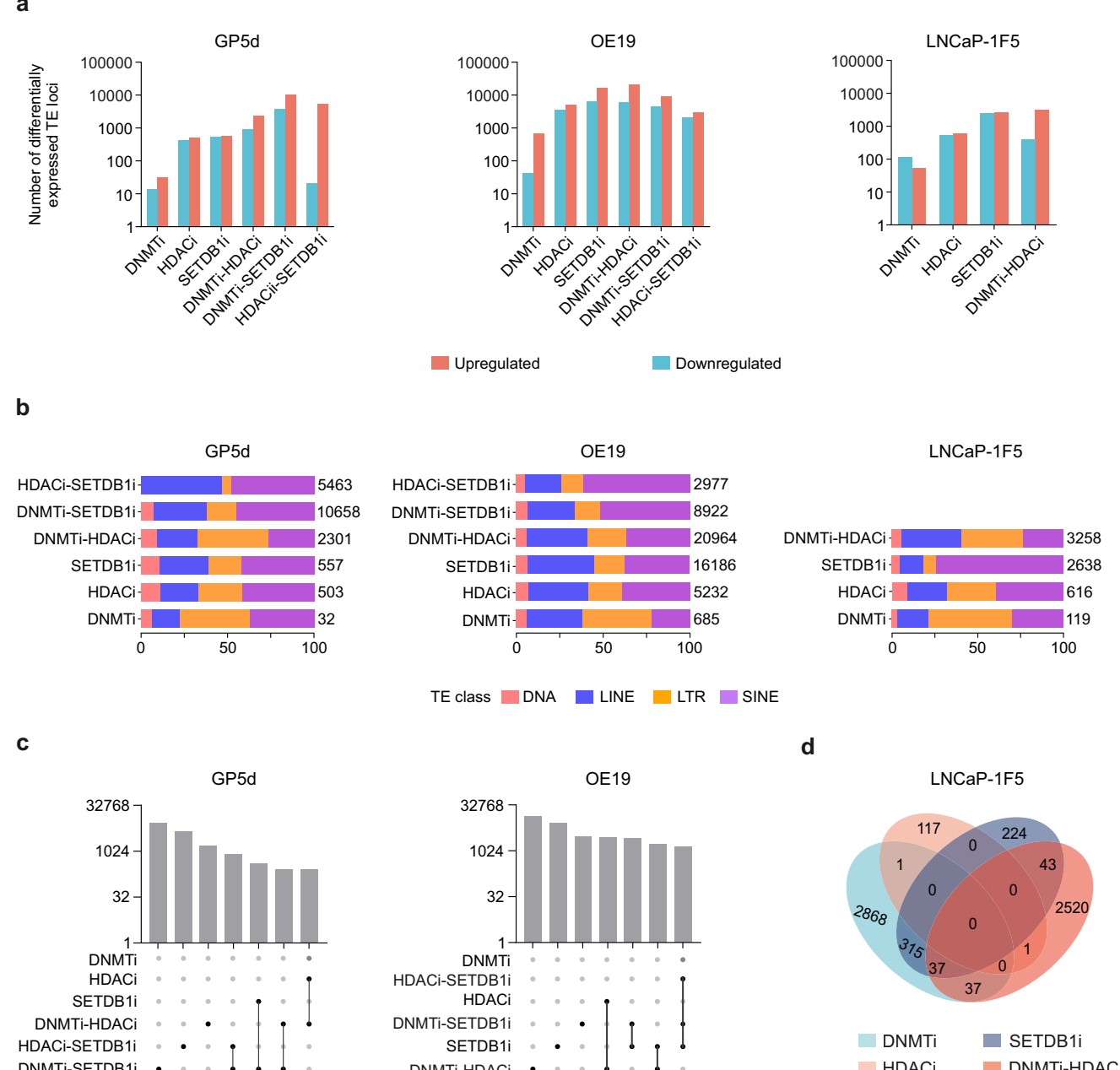

**Fig. 2 | DNMT and HDAC co-inhibition synergistically derepresses TEs.**
**a** Number of differentially expressed TE loci induced by CMEi in GP5d, OE19 and LNCaP-1F5 cells. TE loci that meet the threshold criteria of log2FC > 1.5 and adjusted p < 0.05 compared to vehicle-treated cells are considered as differentially expressed (DESeq2). **b** The proportion of derepressed TE loci by CMEi belonging to major TE classes in GP5d, OE19, and LNCaP-1F5 cells. Derepressed TE loci were labeled by TE class and their counts presented as percentage of total. Numbers represent total derepressed TE loci. **c** Top seven intersecting sets of derepressed individual TE loci from six different CMEi in GP5d and OE19 cells. The bar plot shows the number of derepressed TE loci and the matrix below the bar plot indicates CME inhibition represented by each bar. **d** Overlap of derepressed individual TE loci by CMEi in LNCaP-1F5 cells. Source data are provided as Supplementary Data 11.

LTR subfamilies, such as MER61 and LTR14, with a high proportion of p53 response elements (p53REs) were upregulated in WT GP5d cells in comparison to p53-KO cells (Fig. 3a). CMEi treated p53-KO cells showed distinct effects: for example, LTR12D elements were derepressed by DNMTi, HDACi, and DNMTi-HDACi in p53-KO GP5d cells, whereas only DNMTi-HDACi derepressed LTR12D in WT cells (Figs. 1d and 3b). The loss of p53 potentiated the effect of all different CMEi treatments, as a greater number of TE loci were derepressed in the p53-KO GP5d and LNCaP-1F5 cells compared to WT cells (Figs. 3c, 2a, and Supplementary Fig. 6a). This was confirmed by the reduced effect of DNMTi-HDACi on TE expression in GP5d p53-KO cells upon p53 rescue at both subfamily and locus levels (Fig. 3d). Furthermore, the number of derepressed TE loci by co-inhibition of DNMT and HDAC in GP5d, OE19, and p53-KO GP5d cells showed an inverse correlation with the functional p53 status (Fig. 3e), indicating that the loss of p53 is associated with stronger TE derepression.

Analysis of the effect of DNMTi-HDACi on the expression of p53 itself revealed p53 downregulation at both transcript and protein levels (Fig. 3f and Supplementary Fig. 5a) and reduced H3K27ac enrichment at the *TP53* promoter in GP5d and OE19 cells (Fig. 3g and Supplementary Fig. 5e). H3K4me3 showed a larger reduction in GP5d compared to OE19 cells, consistent with the stronger downregulation of p53 expression in GP5d cells

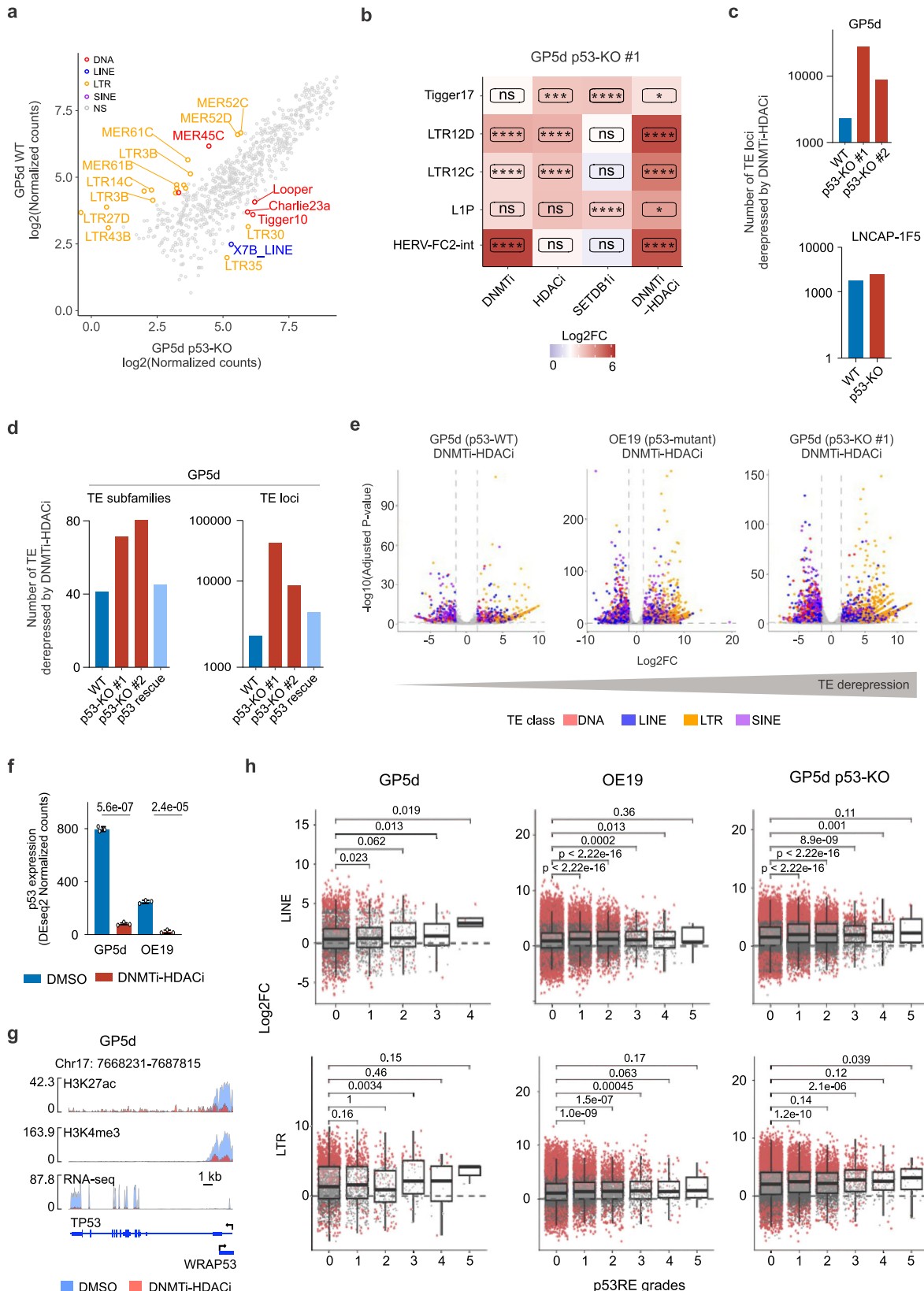

(Fig. 3g and Supplementary Fig. 5e). Of note, reduced p53 expression was associated with an increase in p63 expression, another TF in the p53 family (Supplementary Fig. 5g).

To study the functional consequences of reduced p53 expression upon DNMTi-HDACi, we compared the expression of known p53 target genes

regulated by LTRs harboring p53REs[23], such as *DHX37* and *TMEM12*, that are controlled by LTR10E and LTR10B1 elements, respectively. In GP5d cells, the gene promoters were enriched for H3K27ac and H3K4me3 marks and the LTR elements for p53 and H3K27ac, but DNMTi-HDACi resulted in a loss of H3K27ac signal at the LTR10s and a reduction of H3K4me3

**Fig. 3 | Loss of p53 is associated with a stronger derepression of TEs by DNMT and HDAC co-inhibition. a** Differentially expressed TE subfamilies between GP5d and p53-KO GP5d cells. Scatter plot shows normalized RNA-seq read counts for TE subfamilies. Differentially expressed TE subfamilies are labeled by TE class. **b** Distinct TE subfamilies derepressed by CMEi in GP5d p53-KO cells. Expression changes for TE subfamilies (log2FC) were compared between different CME treatments in GP5d p53-KO cells. Significance symbols: **** indicates p < 0.0001, ***p < 0.001, **p < 0.01, *p < 0.05, ns = non-significant |log2FC| < 1.5 or p > 0.05. **c** Loss of p53 is associated with stronger derepression of TEs by DNMTi-HDACi. Bar plots compare (i) the number of derepressed TE loci by DNMTi-HDACi in WT GP5d and two independent p53-KO clones and (ii) the number of derepressed TEs loci between LNCAP-1F5 p53-WT and p53-KO cells. **d** Rescue experiments showing that p53 reintroduction is associated with weaker derepression of TEs at the subfamily and locus levels by DNMTi-HDACi. Bar plot comparing the number of derepressed TE subfamilies and TE loci by DNMTi-HDACi in GP5d, GP5d p53-KO clones #1 and #2 and p53-transfected GP5d p53-KO clone #1. **e** Volcano plots

showing the differentially expressed individual TE loci by DNMTi-HDACi in GP5d, OE19, and GP5d p53- KO cells. **f** Bar plot comparing p53 expression in GP5d and OE19 cells treated with DNMTi-HDACi vs. DMSO control (unpaired two-sided t-test). **g** Genome browser snapshot of the *TP53* gene locus showing the ChIP-seq signals for H3K27ac and H3K4me3 and an RNA-seq signal track for both DMSO control and DNMTi-HDACi-treated GP5d cells. **h** Comparison of TE expression changes induced by DNMTi-HDACi treatment between TEs harboring p53REs with different strengths. Expressed TEs with p53REs were stratified into five grades from least to most likely p53REs with transactivation potential using p53retriever[34]. Boxplots show the number of up- and downregulated TE loci upon DNMTi-HDACi for each grade, significantly differentially expressed loci are marked with red (Adjusted p < 0.05, |log2FC| > 1). Comparisons between grades were performed with one-sided Wilcoxon tests. Number of LINEs and LTRs loci for each grade are shown in Supplementary Data 4. Source data are provided as Supplementary Data 11.

levels at the promoters commensurate with the downregulation of the genes (Supplementary Fig. 5f).

To further understand the p53-mediated control of TEs, we classified TEs derepressed by DNMTi-HDACi into six groups based on the enrichment of p53REs with different transactivation potentials (Grade 0: no p53RE, Grades 1-5: from least to most likely to transactivate)[34] and compared expression changes induced by DNMTi-HDACi treatment. Majority of the LINES and LTRs in GP5d, OE19 and GP5d p53-KO cells did not have a p53RE (Grade 0) or had a low-affinity p53RE that are unlikely to be functional (Grade 1) (Fig. 3h and Supplementary Data 4). LINEs and LTRs harboring p53REs with stronger transactivation potential (Grades 3-5) showed stronger derepression by DNMTi-HDACi treatment, but they represented only a small fraction of the derepressed TEs (Fig. 5h). In agreement with this functional p53RE enrichment analysis, motif enrichment analysis of TEs derepressed by DNMTi-HDACi in WT and p53-KO GP5d and OE19 cells revealed that most of the derepressed TEs (more than 77%) do not have a consensus p53 motif (Supplementary Fig. 5h), suggesting a DNA binding-independent role of p53 in their repression. Interestingly, LINEs represented a higher percentage of the derepressed TE loci in the p53-KO cells compared to the WT cells (Supplementary Fig. 6b, and Supplementary Data 3) in agreement with higher LINE1 expression in p53-mutant tumors[35,36]. As in WT cells, derepressed TEs after different CMEi treatments in p53-KO cells showed little overlap (Supplementary Fig. 6c).

Collectively, DNMTi-HDACi treatment had a stronger effect on TE derepression in cells without functional p53 compared to WT cells. The majority of derepressed TEs did not have a p53RE or harbor a low-affinity p53RE that are unlikely to be functional, suggesting that the induced changes in TE expression might be due to sequence independent and indirect effects of p53.

### Distinct epigenetic mechanisms govern derepression of LTR12C induced by co-inhibition of DNMT and HDAC

For understanding the effect of DNMTi-HDACi on TE expression, we performed motif enrichment analysis for the genomic sequences of TEs derepressed by DNMTi-HDACi, revealing KRAB-ZFP motifs, such as ZNF460, ZNF135, and ZBTB6 (Fig. 4a). For further analysis of the regulatory mechanisms, we focused on LTR12C elements that are robustly derepressed by DNMTi-HDACi in GP5d[5], OE19 and GP5d-p53 KO cells at both subfamily and locus levels (Figs. 1d and 4b). A total of 499 and 861 LTR12C loci were derepressed in GP5d and OE19 cells, respectively, majority of which (88%) were shared between the two cell lines (Fig. 4c). The loci were enriched with Forkhead family as well as GATA, ERF, and NFY motifs in both cell lines (Supplementary Fig. 7a), in agreement with earlier studies showing NFYA binding associating with transcription initiation from LTR12C[5,18].

Comparison of epigenetic states of derepressed LTR12Cs in GP5d and OE19 cells revealed differences in their epigenetic regulation (Fig. 4d, and

Supplementary Fig. 7b). In OE19 cells, derepressed elements were enriched for ATAC-seq signal and CUT&TAG signals for H3K27me3 and H3K4me1, suggesting a poised enhancer state[37] (Fig. 4d). Furthermore, CUT&TAG signal for RNAPII and Serine-5-phosphorylated (Ser5p) RNAPII showed pausing at LTR12C transcription start sites (TSS) (Fig. 4d). Actively transcribing RNAPII was also suggested by the enrichment of KAS-seq signal[38], a measurement of ssDNA at transcriptionally engaged RNAPII (Fig. 4e). On the contrary, in GP5d cells derepressed LTR12Cs were enriched for ATAC-seq and H3K27me3 but not for H3K4me1 (Supplementary Fig. 7b).

Furthermore, nanopore sequencing analysis of DNMTi-HDACi-treated GP5d cells showed reduced levels of CpG methylation at almost all derepressed TEs, including the derepressed LTR12C elements (Fig. 4f, and Supplementary Fig. 8a). In DNMTi-HDACi-treated GP5d cells, the derepressed LTR12Cs gained active histone marks such as H3K27ac and H3K4me3 (Fig. 4g). In contrast, H3K4me3 was already enriched at the LTR12C loci in untreated OE19 cells (Fig. 4h), and DNMTi-HDACi only increased the H3K27ac signal (Fig. 4h).

Analysis of gene expression changes upon DNMTi-HDACi within ±50 kb of a derepressed LTR12C elements showed 132 and 89 genes differentially expressed in GP5d and OE19 cells, respectively (Fig. 4i and Supplementary Fig. 7c), speaking for their function as active regulatory elements. Representative examples of the target gene expression are shown for the *DDIT4L* gene in GP5d cells and the *PGPEP1L* gene in OE19 cells (Fig. 4j, k; Supplementary Fig. 7d).

In conclusion, our detailed analyses of the regulation of LTR12C elements revealed that reduced CpG methylation along with the gain of H3K27ac determine their activation upon DNMTi-HDACi.

### Inhibition of SETDB1 derepresses cancer cell type-specific TEs

Among the individual CMEi treatments, SETDB1i had a stronger effect on TE expression compared to DNMTi and HDACi treatments alone. Thus, for comprehensive understanding of SETDB1i on TE regulation, we compared TE derepression in GP5d, OE19, LNCaP-1F5, GP5d p53-KO, LNCaP-1F5 p53-KO cells and in SETDB1-KO A375 melanoma cells[15]. LTR elements were overrepresented among the derepressed TEs in all cells (Fig. 5a, and Supplementary Data 5) but there were cancer type-specific differences in the derepressed subfamilies. LTR12C, LTR12D, and LTR12 were derepressed in A375 cells as reported earlier[15], whereas LTR7B, LTR7C, and L1ME3E were derepressed in GP5d cells and LTR51, LTR12C, and MER51E in OE19 cells (Supplementary Fig. 9). Generally, the effect of SETDB1 KO or inhibition on differential TE expression was stronger in p53-mutant OE19 and in p53-KO GP5d and LNCaP-1F5 cells compared to the respective WT cells (Fig. 5b). SINEs such as AluY and AluS subfamilies represented more than 70% of the TEs derepressed by SETDB1i in WT and p53-KO LNCaP-1F5 cells (Fig. 5c and Supplementary Fig. 9), whereas Alus represented only 14% of derepressed TEs in A375 cells (Fig. 5d). In line with this, TF motif enrichment showed differences

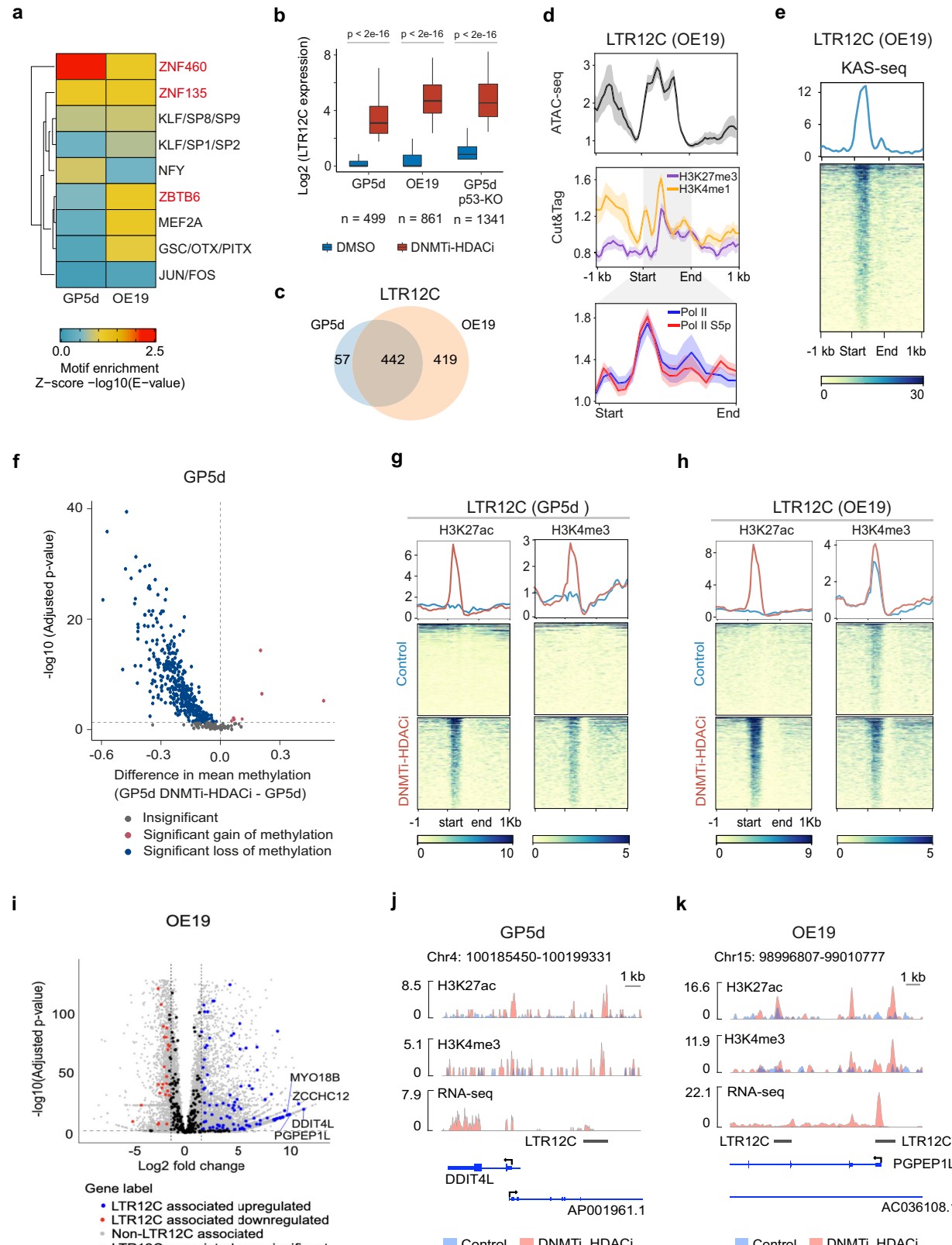

between the cancer types: KRAB-ZFPs such as ZNF460 were enriched in GP5d and A375 cells, whereas ZNF135 motif was exclusively enriched in GP5d cells (Fig. 5e). Collectively, our results demonstrate that SETDB1 KO or inhibition results in cell type-specific derepression of TEs in cancer cell lines.

## Distinct effects of CMEi treatments on accumulation of dsRNAs derived from inverted repeat Alu SINEs

Inverted repeat (IR) Alu elements are a major source of immunogenic dsRNA. To understand the mechanistic effect of epigenetic drugs on immune responses, we characterized CMEi-induced effects on IR-Alu

**Fig. 4 | Distinct epigenetic mechanisms govern the derepression of LTR12C by co-inhibition of DNMT and HDAC. a** TF motif enrichment at TE sequences derepressed by DNMTi-HDACi in GP5d and OE19 cells. After performing motif enrichment analysis for individual motifs, similar motifs were combined into motif clusters from ref. 78. The representative TF clusters are labeled on the right. The minimum E-value found for an individual TF for each motif cluster was plotted in the final figure. **b** Boxplots comparing the expression of derepressed LTR12C in DMSO and DNMTi-HDACi GP5d, OE19 and GP5d-p53-KO cells (two-sided Wilcoxon paired test). The lower and upper hinges of the boxes represent the 25th to 75th percentiles, the midline is the median, and the whiskers extend from the hinges to the minimum and maximum values by 1.5 * interquartile range (IQR). **c**, Venn diagram showing the overlap between LTR12Cs derepressed by DNMTi-HDACi in GP5d and OE19 (shown in Fig. 4b). **d** Metaplots of ATAC-seq and CUT&TAG for H3K27me3, H3K4me1, RNAPII and Ser5p-RNAPII at derepressed LTR12Cs in DNMTi-HDACi OE19 cells (shown in Fig. 4b). Derepressed LTR12C elements show a poised chromatin state in OE19 cells, whereas LTR12C in GP5d cells were enriched with repressive H3K27me3 marks (see Supplementary Fig. 7b). **e** Heatmap showing the KAS-seq signal at derepressed LTR12C elements in DNMTi-HDACi OE19 cells (shown in Fig. 4b). **f** Volcano plot of nanopore sequencing data comparing CpG

methylation levels at derepressed LTR12C elements (n = 499) in control and DNMTi-HDACi GP5d cells (shown in Fig. 4b). Significance was determined with a one-sided Fisher's exact test. *P*-values were corrected with Benjamini-Hochberg method. **g** Heatmaps showing the ChIP-seq signals for H3K27ac and H3K4me3 in control and DNMTi-HDACi GP5d cells (shown in Fig. 4b). **h** Heatmaps showing the ChIP-seq signals for H3K27ac and H3K4me3 in control and DNMTi-HDACi OE19 cells (shown in Fig. 4b). **i** Derepressed LTR12C elements in DNMTi-HDACi GP5d and OE19 cells. Volcano plots show the changes in gene expression for derepressed LTR12C-associated genes for OE19 cells. Analysis of differentially expressed genes in OE19 cells with DNMTi-HDACi revealed significant upregulation of genes in the vicinity of derepressed LTR12C elements (±50 kb). In total, 102 out of 132 and 70 out of 89 differentially expressed genes were upregulated in GP5d and OE19 cells, respectively. **j** Genome browser snapshot of a derepressed LTR12C in control and DNMTi-HDACi GP5d cells. Panels show the ChIP-seq signals for H3K27ac, H3K4me3, and an RNA-seq signal track. **k** Genome browser snapshot of a derepressed LTR12C in control and DNMTi-HDACi OE19 cells. Each panel shows the ChIP-seq signals for H3K27ac, H3K4me3 and an RNA-seq signal track. Source data are provided as Supplementary Data 11.

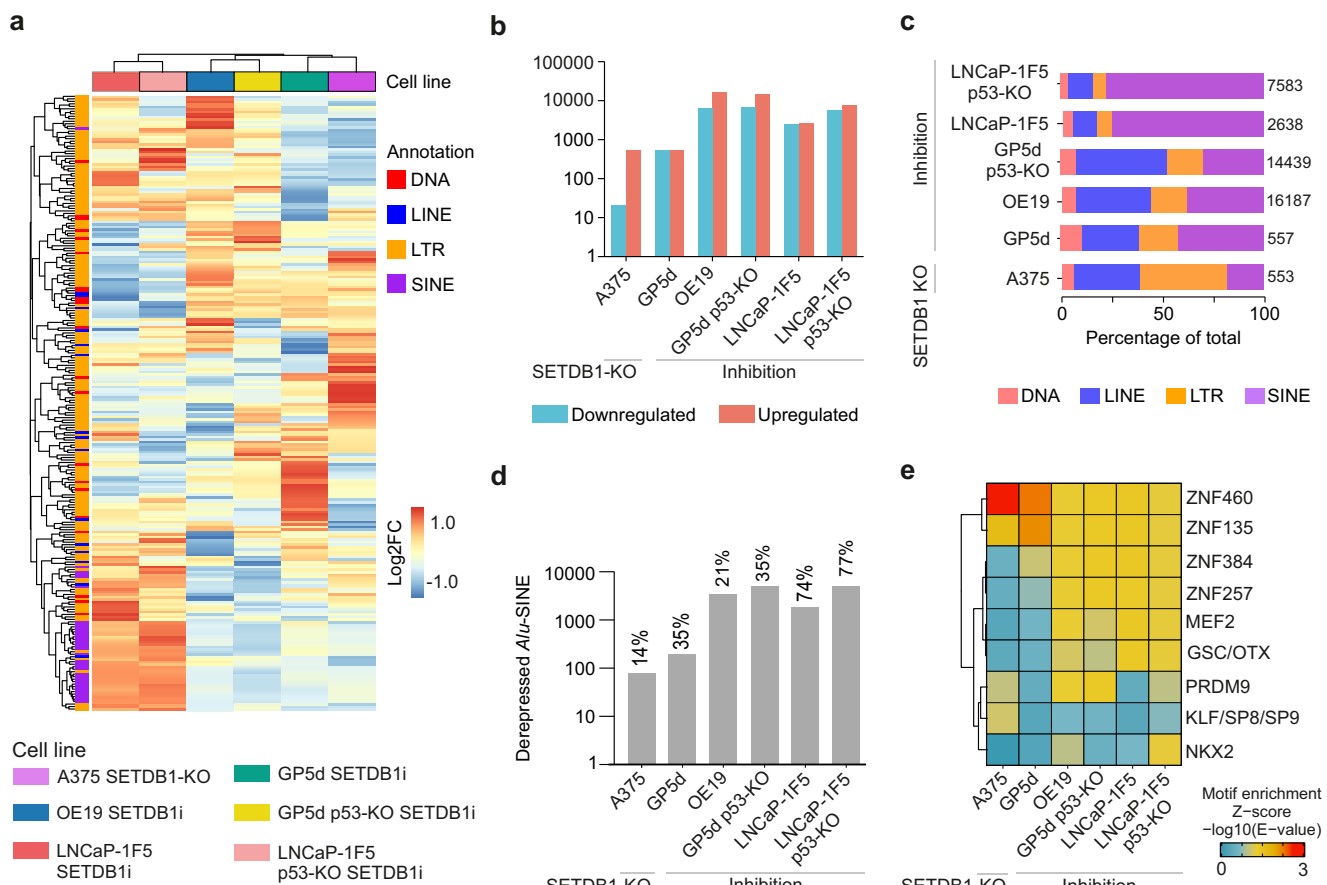

**Fig. 5 | Cell type-specific TEs derepressed by SETDB1 inhibition in cancer cell lines. a** Comparison of SETDB1 KO/inhibition-induced changes in the expression of TE subfamilies between six different cell lines. Differential expression analysis of RNA-seq data was performed with DESeq2. All TE subfamilies with absolute log2FC > 1.5 (treatment vs DMSO or A375 KO vs A375) and adjusted p < 0.05 in at least one CMEi treatment in one cell line were selected and their expression changes in log2FC was plotted. Rows and columns are clustered with hierarchical clustering. **b** Number of differentially expressed TE loci by SETDB1 KO/inhibition in A375, WT and p53-KO GP5d, OE19, and WT and p53-KO LNCaP-1F5

cells. **c** The proportion of TEs derepressed by SETDB1 KO/inhibition belonging to the major TE classes among the derepressed TEs by in A375, WT and p53-KO GP5d, OE19, and WT and p53-KO LNCaP-1F5 cells. **d** Number of Alu SINEs derepressed by SETDB1 KO/inhibition in A375, WT and p53-KO GP5d, OE19, and WT and p53-KO LNCaP-1F5 cells. Number above the bar represents the percentage of Alu SINEs in total derepressed TEs. **e** Motif enrichment analysis for TEs derepressed by SETDB1 KO/inhibition in A375, WT and p53-KO GP5d, OE19, and WT and p53-KO LNCaP-1F5 cells. The analysis was performed as described in Fig. 4a. Source data are provided as Supplementary Data 11.

elements. Out of all derepressed TE loci in OE19, GP5d, and GP5d p53-KO cells, SINEs represented 37%, 26%, and 25%, respectively, and most of the derepressed SINEs were Alu elements (see Fig. 2b, e). To characterize transcriptional and epigenetic changes at IR-Alu elements by DNMTi-

HDACi, we selected the transcriptionally active IR-Alu elements from all IR-Alu elements in the human genome (see "Methods" for details), resulting in a total of 4966, 83952, and 84606 active IR-Alus in GP5d, OE19, and GP5d p53-KO cells, respectively. Differential expression analysis revealed

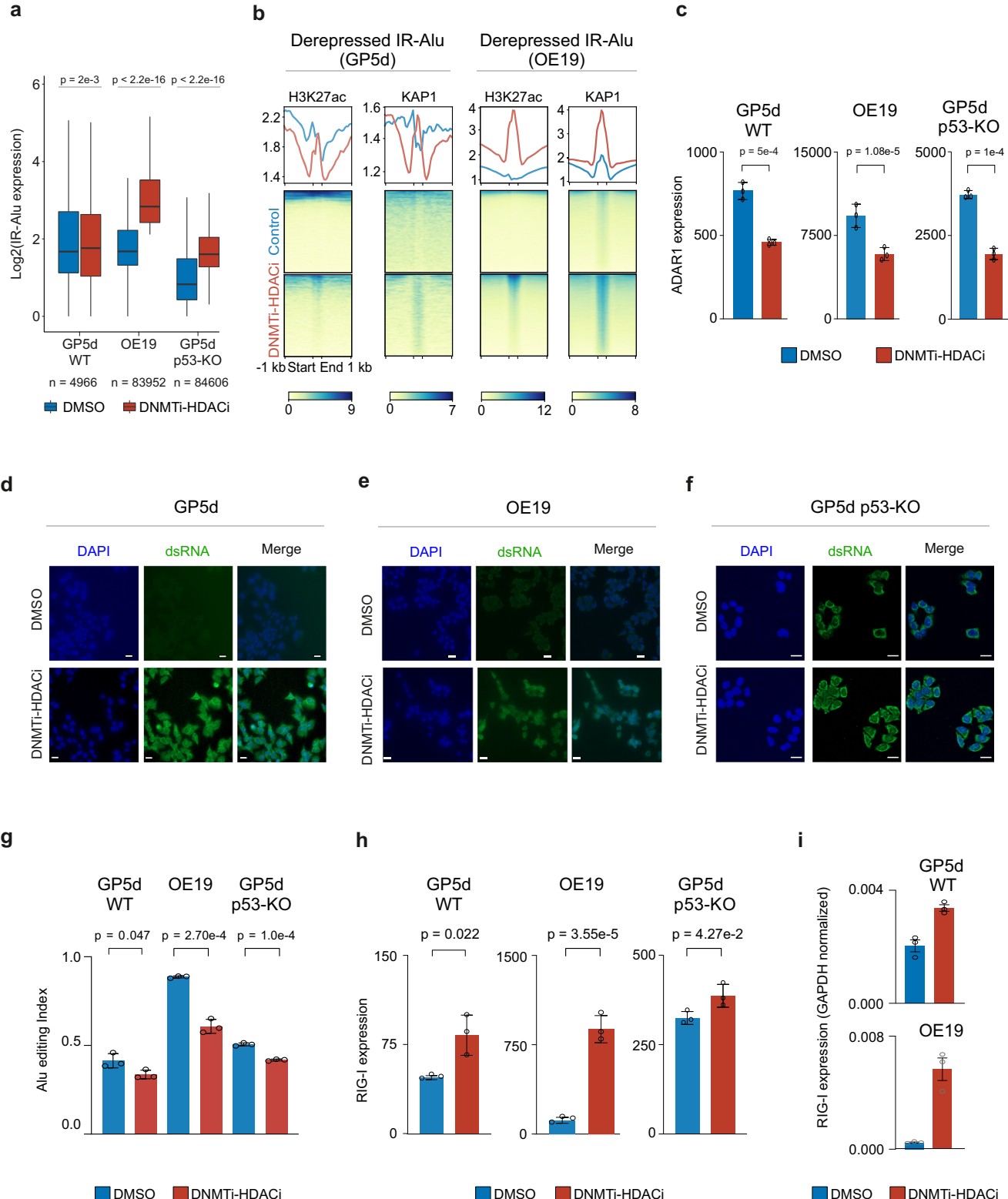

that the majority of transcriptionally active IR-Alus were upregulated upon DNMTi-HDACi in all three cell lines (Fig. 6a), but the derepression was considerably stronger (at least 18-fold higher) in OE19 and GP5d p53-KO cells compared to GP5d cells. Increased H3K27ac ChIP-seq signal was observed at the derepressed IR-Alus in GP5d and OE19 cells (Fig. 6b). OE19 cells showed stronger gain for H3K27ac ChIP-seq signal, consistent with a stronger increase in IR-Alu expression (Fig. 6a, b). Derepressed IR-Alu

elements in the three cell lines cells revealed both common and distinct TF motif patterns (Supplementary Fig. 10a), suggesting a cell type-specific *cis*-regulatory logic at distinct genomic locations.

IR-Alu elements form dsRNA structures that are post-transcriptionally modified by ADAR1[39] through adenosine to inosine conversion (A-to-I editing)[40], and depletion of ADAR1 from GP5d cells led to increased accumulation of cytoplasmic dsRNA (Supplementary Fig. 10b, c). Next, we

**Fig. 6 | Inverted repeat Alu SINEs are derepressed by co-inhibition of DNMT and HDAC in GP5d and OE19 cells. a** Boxplots showing a comparison of expression of transcriptionally active IR-Alu SINEs (total sum of RNA-seq reads for DMSO and DNMTi-HDACi ≥ 5, see "Methods" for details) in DMSO and DNMTi-HDACi GP5d, OE19, and GP5d p53-KO cells. Majority of IR-Alu SINEs are derepressed by DNMTi-HDACi (two-sided Wilcoxon paired test). The lower and upper hinges of the boxes represent the 25th to 75th percentiles, the midline is the median, and the whiskers extend from the hinges to the minimum and maximum values by 1.5 * IQR. **b** Heatmap showing the ChIP-seq signal for H3K27ac at IR-Alu SINEs in DMSO and DNMTi-HDACi GP5d and OE19 cells. **c** Comparison of normalized RNA-seq read counts for ADAR1 gene in DMSO and DNMTi-HDACi GP5d, OE19, and GP5d p53-KO cells. The graph shows mean ± SD values for three biological replicates (two-sided unpaired t-test). **d** Microscopy images for GP5d cells treated with DMSO or DNMTi-HDACi. DNA was stained with DAPI (blue), and dsRNA was stained using the J2 antibody (green). All scale bars are 20 μm. Cytoplasmic levels of dsRNA increased in DNMTi-HDACi treated GP5d cells as compared to the DMSO. **e** Microscopy images for dsRNA staining in OE19 cells treated with DMSO or DNMTi-HDACi, as shown in Fig. 6d. **f** Microscopy images for dsRNA staining in GP5d p53-KO cells treated with DMSO or DNMTi-HDACi, as shown in Fig. 6d. **g** Alu editing index (AEI) was calculated by using RNAeditingIndexer[42] tool on RNA-seq data. Bar plots showing AEI for DMSO and DNMTi-HDACi GP5d, OE19, and GP5d p53-KO cells. The graph shows mean ± SD values for three biological replicates (two-sided unpaired t-test). **h** Bar plots comparing *RIG1* gene expression in DMSO and DNMTi-HDACi GP5d, OE19, and GP5d p53-KO cells. The graph shows mean ± SD values for three biological replicates (two-sided unpaired t-test). **i** qRT-PCR data showing RIG-I mRNA expression in GP5d and OE19 cells treated with DMSO and DNMTi-HDACi (GAPDH normalized). The graph shows mean ± SD values for three biological replicates. Source data are provided as Supplementary Data 11.

analyzed ADAR1 expression upon distinct CMEi treatments and their effect on dsRNA accumulation. In agreement with earlier studies[21,41], we observed an increased expression of ADAR1 in DNMTi-treated p53 wild-type GP5d cells (Supplementary Fig. 10d). Interestingly, ADAR1 mRNA and protein expression was downregulated by DNMTi-HDACi in both GP5d and OE19 cells, despite the differences in the baseline ADAR1 expression levels in these cell lines (Fig. 6c. Supplementary Figs. 10d, 11a, b). Of note, GP5d cells harbor a nonsense mutation in one of the ADAR1 alleles (Supplementary Data 7), potentially explaining its low baseline expression level. SETDB1i increased ADAR1 mRNA expression particularly in GP5d cells along with upregulation of its p110 protein isoform (Supplementary Figs. 10d, 11a, b). Both isoforms have a higher baseline expression level in OE19 cells, and thus the effect of SETDB1i on p110 upregulation was weaker, whereas the level of p150 isoform decreased. In line with ADAR1 downregulation, DNMTi-HDACi induced the accumulation of cytoplasmic dsRNA while DNMTi and SETDB1i had a weaker effect (Fig. 6d–f, and Supplementary Figs. 10d–f and 11c).

Commensurate with the changes in ADAR1 expression, we observed a significant decrease in Alu editing index (AEI)[42] by DNMTi-HDACi but an increase after SETDB1i (Fig. 6g, and Supplementary Fig. 12a). Notably, the increase in AEI was at least two-fold higher in OE19 cells compared to WT and p53-KO GP5d cells. Interestingly, DNMTi-SETDB1i and HDACi-SETDB1i also led to an increase in AEI in GP5d and OE19 cells, the effect was stronger in p53-WT GP5d cells (Supplementary Fig. 12a). The genomic distribution of Alu RNA editing events showed distinct patterns in the five cell lines (Supplementary Fig. 12b). In GP5d cells, two thirds of the Alu editing-induced mismatch sites were observed within the exonic Alu elements with similar genomic distribution between control and DNMTi-HDACi cells (Supplementary Fig. 12b). In OE19 and GP5d p53-KO cells, intronic Alus represented 29% and 35% of the editing events in control cells, and the proportion increased to 42% and 44% in DNMTi-HDACi cells, respectively, while editing in exonic regions decreased (Supplementary Fig. 12b).

Consistently with the accumulation of dsRNA in response to DNMTi-HDACi treatment, the expression of RIG-I inflammasome, a known cytoplasmic dsRNA sensor[43], significantly increased in GP5d, OE19, and GP5d p53-KO cells upon DNMTi-HDACi (Fig. 6h, i). However, another dsRNA sensor, MDA5 (IFITH1)[44], was downregulated at mRNA but not at protein level in DNMTi-HDACi treated cells (Supplementary Fig. 13a, b). This suggests non-redundant roles for RIG-1 and MDA5 in these cell lines as reported previously in the context of different viral infections[45], potentially due to dsRNA structures, post-translational protein modifications, or differential co-factors[44], warranting further mechanistic studies. Collectively, our results show that different CMEi treatments lead to distinct responses in ADAR1 expression, dsRNA accumulation and Alu RNA editing.

## SETDB1i and DNMTi-HDACi increase TE-chimeric transcript expression and activate an inflammatory response

In addition to dsRNAs, chimeric transcripts derived from TE elements contribute to immune responses induced by TEs[17,18,46]. To study the effect of CMEi on TE-derived chimeric transcripts, we performed TEProf2[16] analysis on RNA-seq data. In agreement with TEs derepression, the expression of TE-chimeric transcripts was strongly affected by functional p53 status: in SETDB1i and DNMTi-HDACi treated cells, over a five times larger number of TEs contributed to chimeric transcript formation in p53-KO GP5d and p53-mutant OE19 cells compared to WT GP5d cells (Fig. 7a, c, Supplementary Fig. 14a, Supplementary Data 6). DNMTi-HDACi significantly increased the expression of TE-chimeric transcripts in all three cell lines (Fig. 7c), whereas SETDB1i only in OE19 and GP5d p53-KO cells (Fig. 7a) and DNMTi in GP5d p53-KO cells in contrast to reduced chimeric expression by HDACi in GP5d cells (Supplementary Fig. 14a).

In agreement with earlier reports[15], most of the TE-chimeric transcripts were derived from LTR12C (Fig. 7d), but we also detected the contribution of other cell type-specific TEs. In response to SETDB1i, THE1B LTRs in A375 cells, L1PB LINE in GP5d p53-KO cells, and L2a, L1PA3, L1Hs LINEs as well as AluJb and AluSp SINEs in OE19 and GP5d p53-KO cells formed chimeric transcripts (Fig. 7b). Similarly, DNMTi-HDACi induced chimeric transcripts from L1PA2, L2a, and AluSx subfamilies in OE19 and AluY and LTR12D in GP5d-KO cells (Fig. 7d). Genome browser snapshots for representative examples include TE-chimeric transcripts formed by an intronic Alu with exons of *GID8* and *YPEL5* genes in SETDB1i GP5d and OE19 cells, respectively, and LTR12-derived transcripts formed with *PARP16* gene in DNMTi-HDACi GP5d p53-KO cells and *FBP2* gene in OE19 cells (Supplementary Fig. 14b, c). In agreement with increased Alu editing in the absence of p53, more Alu-derived transcripts were induced in OE19 and GP5d p53-KO cells compared to WT GP5d cells (Fig. 7d). Collectively, we observed an inverse correlation between functional p53 activity and CMEi-induced TE-chimeric transcripts.

Finally, we analyzed the effect of CMEi on global gene expression profiles. DNMTi-HDACi had the strongest effect on gene expression in all five cell lines compared to individual CMEi treatments, and loss of p53 was associated with stronger changes in gene expression compared to p53 WT cells (Supplementary Fig. 15a). Gene set enrichment analysis of the differentially expressed genes induced by DNMTi-HDACi revealed the enrichment of an inflammatory response pathway in both GP5d and GP5d p53-KO cells (Supplementary Fig. 16a). Similarly, SETDB1i resulted in the enrichment of the inflammatory response and interferon-alpha (IFNα) signaling pathways in GP5d, GP5d p53-KO, and OE19 cells (Supplementary Fig. 16b). Both DNMTi-HDACi and SETDB1i treatments induced the expression of IFN-stimulated genes (ISGs)[47] in p53 wild-type and mutant cells (Fig. 7e, g, and Supplementary Figs. 16c, 17a), such as IFNα (Fig. 7g) and phosphorylated IRF7 (phospho-IRF7) (Fig. 7f) indicating activation of type I IFN signaling pathway[48]. DNMTi-HDACi induced expression of IRF7 in all three cell lines. However, an increase in phospho-IRF7 levels was stronger in GP5d WT and p53-KO cells compared to OE19 cells (Fig. 7f). Collectively, these gene expression changes together with increased levels of phospho-IRF7 protein show that DNMTi-HDACi and SETDB1i activate a strong interferon response in cancer cells through TE derepression in both p53 WT and mutant cancer cells.

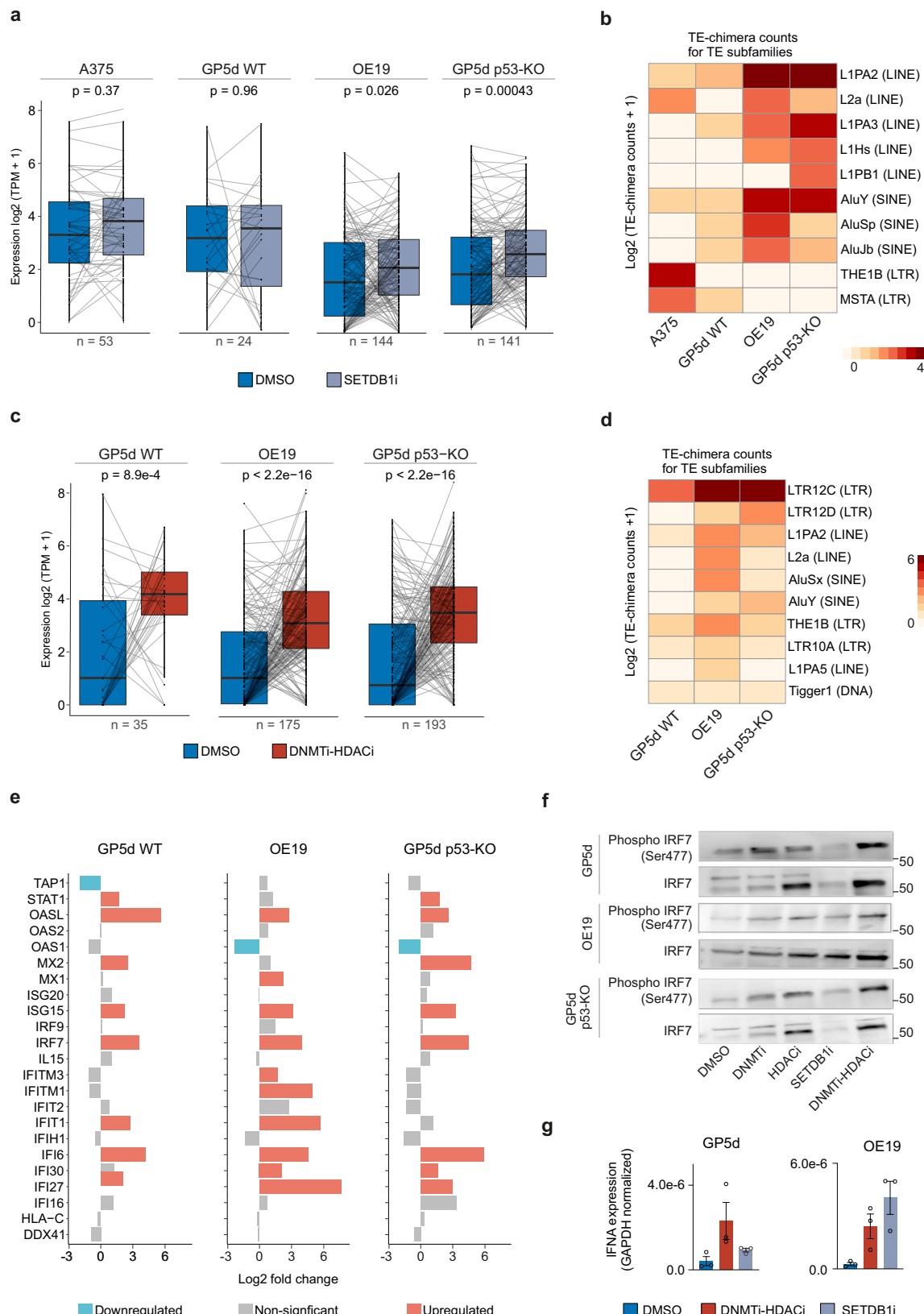

Cell type-specificity of the TEs derepressed by DNMTi-HDACi in GP5d and OE19 cells prompted us to delineate the chromatin state of these derepressed TEs in human cancer tissues. For this, we analyzed the ATAC-seq data for colon adenocarcinoma (COAD) and esophageal carcinoma (ESCA) using The Cancer Genome Atlas (TCGA) datasets[49]. Importantly,

L1PA subfamilies that were specifically derepressed in OE19 and GP5d p53-KO cells were enriched within open chromatin regions in ESCA and COAD patient samples, respectively (Supplementary Fig. 18a). LTR12C and LTR12D elements that were commonly derepressed in all three cell lines were found in both COAD and ESCA patient samples (Supplementary

**Fig. 7 | SETDB1i and DNMTi-HDACi increases TE-chimeric transcript expression and activates inflammatory response. a** SETDB1i/KO increases the expression of TE-chimeric transcripts. Boxplots show the expression of TE-chimeric transcripts in A375, GP5d, OE19, and GP5d p53-KO cells with and without SETDB1i/KO. Expression of TE-chimeric transcripts was analyzed with the TEprof2 pipeline[16]. *P*-values were calculated with a two-sided Wilcoxon test (n = 53, 24, 144, and 141 differentially expressed TE-chimeric transcripts in A375, GP5d, GP5d p53-KO, and OE19, respectively). The lower and upper hinges of the boxes represent the 25th to 75th percentiles, the midline is the median, and the whiskers extend from the hinges to the minimum and maximum values by 1.5 * IQR. **b** Analysis of TE subfamilies from which the TE-chimeric transcripts are derived from upon SETDB1i/KO. The counts for TE-chimeric transcripts are log2-transformed. **c** DNMTi-HDACi increases the expression of TE-chimeric transcripts. Boxplots show the expression of TE-chimeric transcripts in DMSO and DNMTi-HDACi

GP5d, GP5d p53-KO, and OE19 cells. *P*-values were calculated with a two-sided Wilcoxon test (n = 35, 175, and 193 for GP5d, OE19, and GP5d p53-KO cells, respectively). The boxplot features are as in Fig. 7a. **d** Cell type-specific expression of TE subfamilies forming TE-chimeric transcripts by DNMTi-HDACi in cancer cell lines. The counts for TE-chimeric transcripts are log2-transformed. **e** ISGs are upregulated by DNMTi-HDACi. ISG log2FCs are plotted for DNMTi-HDACi treated GP5d, OE19 and GP5d p53-KO cells. **f** SETDB1i and DNMTi-HDACi increased levels of Serine 477 phosphorylated IRF7. Western blot compares Ser477-phospho-IRF7 and total IRF7 protein levels in GP5d, OE19 and GP5d p53-KO cells treated with different CME inhibitors. **g** qRT-PCR data showing IFNα mRNA expression in GP5d and OE19 cells treated with DMSO, DNMTi-HDACi, and SETDB1i (GAPDH normalized). ISGs are upregulated by DNMTi-HDACi and SETDB1i in GP5d and OE19 cells. The graph shows mean ± SD values for three biological replicates. Source data are provided as Supplementary Data 11.

Fig. 18a). Derepressed TEs overlapping with open chromatin regions in patient samples showed enrichment for distinct TF motifs, such as NFY, REST, KLF, and SP families in GP5d cells and ZNF382, ZNF8, ZNF354A, and ZNF136 in OE19 and GP5d p53-KO cells (Supplementary Fig. 18b), suggesting their controlling TFs. Collectively, our results from DNMTi-HDACi-induced cell type-specific derepression of TEs shows association with open chromatin regions in TCGA patient tumor samples.

## Discussion

Understanding the epigenetic regulation of TEs is pertinent for the effective use of epigenetic therapy alone or as an adjuvant to immunotherapy, which have provided promising results in cancer patients[12,13,20]. TE derepression induced by epigenetic therapy plays a key role in sensitizing the cancer cells to immunotherapy[13,17]. While earlier studies have demonstrated TE derepression by few combinations of epigenetic drugs in individual cancer types[15,17,18,21,50], a comprehensive understanding of the mechanisms of TE activation across different cancer types—especially in the context of p53 activity, has remained elusive. To address this, we have systematically targeted three key CMEs—DNMT, HDAC, and SETDB1—using single and combinatorial inhibitor treatments in three cancer types of endodermal origin, each with distinct functional status of p53. We show that different CMEi treatments activate TE expression and TE-derived chimeric transcripts in a cancer type-specific manner, leading to immunogenic response (Supplementary Data 10).

CMEi resulted in derepression of both cell type-specific and common TE subfamilies across three cancer types, DNMTi-HDACi resulting in the most robust and consistent TE activation. At the locus level, TEs induced by different CMEi treatments showed minimal overlap, indicating distinct epigenetic silencing mechanisms due to diverse sequence compositions[51] and genomic locations of different TEs. Mutations in CMEs could contribute to different responsiveness of the three cell lines to CME inhibition, though only missense mutations have been reported in DNMT3A, DNMT3B, HDAC1, HDAC3, and SETDB1 in GP5d and LNCaP cells [https://depmap.org/portal] (Supplementary Data 7). GP5d and LNCaP cells also harbor a non-sense mutation in one of the HDAC4 alleles [https://depmap.org/portal] (Supplementary Data 7), but HDACi treatment strongly induced TE expression also in these cell lines, since several HDAC enzymes are inhibited by the pan-HDAC inhibitor SB939[31]. The specificity of each treatment was confirmed by changes in CME protein levels or corresponding epigenetic modifications, such as increased histone acetylation with HDACi. DAC had only a moderate effect on DNMT1 protein levels, reflecting its function as a nucleotide analog that induces replication-dependent DNA hypomethylation by trapping DNMT1 but does not inhibit its de novo synthesis[52]. However, the effect of DAC treatment on DNA methylation directly at the TE loci was confirmed using long-read nanopore sequencing (see Fig. 4f and Supplementary Fig. 8).

In addition to activation of TEs upon CMEi, we also observed repression of TEs and genes in all five cell lines, particularly by SETDB1i (see Fig. 2a, d, and Supplementary Fig. 15a). Intriguingly, SETDB1i increased H3K9me3 levels in GP5d and OE19 cells (see Supplementary

Fig. 2a), but with concomitant increase in the expression of SUV39H2 and strong decrease in KDM4A-C expression (Supplementary Fig. 15b). This suggests that effect of SETDB1i on H3K9me3 regulation is complex and the net effect on H3K9me3 involves multiple enzymes, such as the interplay between different histone methyltransferases (SETDB1 and SUV39H family of proteins) and lysine-specific demethylases (KDM4s)[53]. This warrants further studies to understand the complex dynamics of H3K9me3 regulation.

We found that at the concentrations used in this study, SETDB1i had a stronger effect on TE expression compared to DNMTi and HDACi treatments alone. Of note, we did not observe stronger TE derepression by DNMTi compared to HDACi. Combination treatment for DNMTi-HDACi showed a synergistic effect on TE derepression in all cell lines in agreement with earlier reports[17,18], but in colon cancer cells the effect was even stronger with DNMTi-SETDB1i and HDACi-SETDB1i combinations which has not been reported earlier. Interestingly, epigenetic states of derepressed LTR12C elements also showed cell type-specificity in cancer cell lines, such as increased H3K4me3 signals upon DNMTi-HDACi in GP5d that remained stable in OE19 cells. However, the gain of H3K27ac was the major determinant of the transcriptional activation of TEs in all cell lines.

p53 has been described both as a transcriptional activator and a repressor of protein-coding genes[23,54], and our results on the TE regulation by p53 is concordant. Importantly, the role of p53 in TE regulation under epigenetic therapies has not been systematically studied despite the prevalence of p53 mutations in human cancers. Our results show that loss of p53 activity results in a stronger TE derepression by all different CMEi treatments. This effect can be rescued by re-introducing p53 that resulted in reduced number of derepressed TEs, supporting the repressive function of p53 in TE regulation. Intriguingly, majority of the derepressed TEs lack a consensus p53 motif, and only very few TEs that harbor p53REs with strong transactivation potential[34] were more frequently upregulated by DNMTi-HDACi. This agrees with previous studies reporting that only few specific TE families, namely LTR10 and MER61, are enriched with p53REs and most other ERVs and TE families lack p53 sites[23]. Consistent with this, we also observed MER61 upregulation in WT GP5d cells compared to p53-KO cells, supporting direct activation of these elements by p53. Repression of LINE1-elements through direct p53 binding has been reported at distinct L1 elements[22], but multiple previous studies support the notion that the repressive effects of p53 are largely indirect, driven by downstream effectors such as p21, E2F7 or miRNAs, and that only transactivation is induced through direct p53 binding (reviewed in ref. 55). Our results demonstrated a major role for p53 in TE derepression upon CMEi treatments through indirect mechanisms, suggesting either a sequence-independent DNA binding of p53 through CTD[38] or indirect genomic control of these elements, warranting further investigation given the complexity involved in p53-mediated repression[55].

In line with previous reports[20,21], we show that CMEi leads to expression of IR-Alu SINEs, a major source of immunogenic dsRNA. ADAR1 enzymes edit these dsRNAs to dampen the immunogenic response[39,40,42], and DNMTi has previously been shown to increase ADAR1 expression[21,41].

To our knowledge, the effect of other CMEi treatments on ADAR1 expression and Alu editing has not reported earlier. Here, our results revealed that DNMTi-HDACi treatment leads to down-regulation of ADAR1 expression, commensurate with reduced Alu RNA editing and accumulation of cytoplasmic dsRNA. On the other hand, SETDB1i-induced ADAR1 expression led to increased Alu editing, which was also seen in our combination treatments with DNMTi-SETDB1i and HDACi-SETDB1i (Supplementary Data 10), suggesting distinct routes of immune response activation by CMEi. Based on our findings, DNMTi-HDACi could be particularly beneficial in cancer immunotherapy, since it robustly induces the expression of immunogenic TEs and simultaneously downregulates ADAR1, the inhibitor of immunogenic dsRNAs. These results provide mechanistic insights for effective immunotherapy such as turning immunotherapy-resistant "cold tumors" like prostate cancer[56] hot by choosing a rational combination of epigenetic drugs, but further in vivo studies are needed to understand the extent of the effects of epigenetic therapy on immune cells in the tumor microenvironment.

Interestingly, SETDB1i had distinct effects on the two protein isoforms of ADAR1 by upregulating p110 and downregulating p150. Previously, it has been reported that more than half of the A-to-I edit sites are selectively edited by the p150 isoform, and the rest can be edited by either p150 or p110[57]. These observations suggest a complex regulation of ADAR1 function by SETDB1i, warranting further mechanistic studies in different cell types. In contrast, DNMTi-HDACi downregulated both ADAR1 isoforms. While the exact mechanism for ADAR1 downregulation remains unclear, previous research has shown that the β-transducin repeat-containing protein (BTRC) can promote degradation of the p110-isoform in response to IFN signaling[58]. Thus, one potential mechanism might be DNMTi-HDACi-induced ADAR1 degradation through BTCR. However, further studies are needed to fully understand how different CMEi treatments regulate ADAR1 expression and stability.

In addition to TE-derived dsRNAs, the chimeric transcripts from TEs can also serve as cancer cell-specific neoantigens that can be exploited for immunotherapy[16,17,46,59]. We observed a strong induction in the expression of TE-chimeric transcripts in response to DNMTi-HDACi in colon and esophageal cancer cell lines in line with earlier findings in lung, colon and CML cancer cells[17,18]. Moreover, we show that SETDB1i also induces TE-chimeric transcripts particularly in p53-deficient colon and esophageal cancer cells, which to our knowledge has not been reported earlier. We also found that different cancer types utilize both common and distinct TEs in chimeric transcript formation upon different CMEi treatments, suggesting that TE-chimeric transcripts can provide cancer-type specific immunogenic signatures for personalized medicine.

In conclusion, we systematically studied the epigenetic regulation of TEs in colon, esophageal, and prostate cancers in the context of p53. Our results show that p53 loss leads to stronger TE activation and TE-chimeric transcript expression independent of the p53 direct DNA-binding activity. Derepressed TEs gain epigenetic signatures of active enhancers and trans-activate nearby genes, eliciting an immune response. Our results show that DNMTi-HDACi combinatorial treatment simultaneously activates immunogenic TEs and downregulates immune response inhibitors, suggesting its potential for improving immunotherapy responses or turning "cold tumors" hot. Collectively, our systematic analyses provide insights for the strategic use of epigenetic therapies in distinct cancer types.

## Methods
### Data acquisition
All sequencing data and download links for annotation files used in this study are listed in Supplementary Data 8 including the relevant references and GEO/ENCODE accessions.

A gene annotation GTF file was downloaded from GENCODE Release 36 for the reference chromosomes. The GTF file was converted into a BED file and TSS and gene body BED files were created with a script adapted from ref. 60 A repeatMasker.txt (2021-09-03) file was downloaded from the UCSC table browser. Only transposable element-derived repeat classes

(LINE, SINE, LTR, and DNA) were retained and a file in BED format was created from the table, totaling 4745258 annotated repeats[61]. LTR12C consensus sequence was downloaded from RepBase[62].

GRCh38 chromosome sizes file was downloaded from UCSC.

GRCh38 blacklist BED file ("ENCFF356LFX [https://www.encodeproject.org/files/ENCFF356LFX/]", release 2020-05-05) was acquired from the ENCODE project.

A genome index was created with bowtie2-build, with chr1-22, X, Y and M fasta files. Alternative, unlocalized and unplaced alternative loci scaffolds were discarded in indexing.

Transcription factor motifs were acquired from JASPAR 2022 CORE non-redundant vertebrate annotations[63]. The position weight matrices in MEME format were used for motif enrichment analyses.

TCGA cancer-type specific ATAC-seq peaks for colon adenocarcinoma (COAD) and esophageal carcinoma (ESCA) were acquired from ref. 49.

### Cell culture
GP5d (Sigma, 95090715) and GP5d p53-KO cells were cultured in DMEM (Gibco, 11960-085) supplemented with 10% FBS (Gibco, 10270106), 2 mM L-glutamine (Gibco, 25030024) and 1% penicillin-streptomycin (Gibco, 15140122). OE19 (Sigma, 96071721) and LNCaP-1F5 cells in RPMI (Gibco, 31870) supplemented with 10% FBS (Gibco, 10270106), 2 mM L-glutamine (Gibco, 25030024) and 1% penicillin-streptomycin (Gibco, 15140122).

GP5d and OE19 cell lines were directly obtained from Sigma, LNCaP-1F5 cells[64] were already available in the lab and low-passage cells were used in all experiments. All cell lines tested negative for mycoplasma contamination upon purchase and were routinely checked as per standard good laboratory practice.

### Cell viability assay for the CME inhibitors
To investigate the effect of each CME inhibitor on cell viability in GP5d and OE19 cells, we performed cell viability assay to calculate dose-response curves for each drug. Cells were seeded in 24 well plates (50000 cells per well) and treated with varying dose (10nM-1000mM) of DAC (72 h), SB939 (24 h) and Mitramycin A (24 h). For DNMTi treatment, media containing DAC was replenished every day. The cell viability was determined by using LUNA-FX7 automated cell counter with Erythrosin B Stain (Logos Biosystems, L13002). Percentage cell viability was calculated for each drug treatment to generate dose response curve.

### Generation of p53-KO GP5d and LNCaP-1F5 cell line by genome editing
The p53-KO GP5d and LNCaP-1F5 cell lines were generated by CRISPR-Cas9 targeting exon 4 of the p53 gene using Alt-R CRISPR-Cas9 from Integrated DNA Technologies. Briefly, Equimolar ratios of target-specific crRNA (Supplementary Data 9) and ATTO550-tracrRNA (IDT, 1075928) were annealed and RNP complex were constituted from Alt-R CRISPR-Cas9 (IDT, 1081060; 1000 ng per 200,000 cells) and target-specific sgRNA (250 ng per 200,000 cells). RNP complex transfected to early passage GP5d and LNCaP-1F5 cells by using CRISPRMAX (Life Technologies, CMAX000003) according to manufacturer's protocol. The next day, ATTO550+ cells were FACS sorted, and single-cell colonies were cultured to produce a clonal p53 KO cell line (Supplementary Fig. 21a). The clonal cells lines were screened for p53 deletion and clones were verified by Sanger sequencing using primers flanking deletion site (Supplementary Data 9).

### ChIP-seq and Nanopore sequencing
ChIP-seq was performed as previously described[65] by using the following antibodies: H3K27ac (Diagenode, C15410196), H3K4me3 (Diagenode, C15410003). Each ChIP-seq reaction was performed by using 2 µg of antibody. In brief, GP5d and OE19 cells were cross-linked for 10 min at room temperature by using formaldehyde (Sigma, F8775). Sonicated chromatin was centrifuged, and the supernatant was used to immunoprecipitated DNA using Dynal-bead coupled antibodies. Immunoprecipitated

DNA was purified and used for ChIP-seq library for Illumina sequencing. ChIP-seq libraries were single-read sequenced on NovaSeq 6000.

For profiling CpG methylation in GP5d cells co-inhibited with DNMT and HDAC, we performed Nanopore sequencing using NaNOMe-seq as described earlier[5]. In brief, DNMTi-HDACi treated GP5d cell nuclei were isolated and treated with GC methylase M.CviPI (New England Biolabs, M0227) as described ref. [66]. Following GC methylation, DNA was isolated from nuclei by using phenol-chloroform extraction protocol, and sequencing library was prepared by using ligation sequencing kit (SQK-LSK109) according to manufacturer's protocol. Total 10 fmol of adaptor-ligated genomic DNA was loaded to the flow cell for sequencing.

## CMEs inhibitor treatment and RNA-seq
DNMT inhibition in GP5d, OE19, LNCaP-1F5, and GP5d p53-KO and LNCaP-1F5 p53-KO cells were performed as described earlier[18]. Cells were seeded in 6 well plate and treated with 500 nM/L DAC (MedChemExpress, HY-A0004). Media containing DAC was replenished daily for 3 days. Cells were harvested after 72 h for RNA isolation.

HDAC inhibition in the five cell lines was performed as described earlier[18]. Cells were treated with 500 nM/L SB939 (MedChemExpress, HY-13322) for 18 h and harvested for RNA isolation.

SETDB1 inhibition in the five cell lines was performed as previously described[32]. Cells were treated with 500 nM/L Mitramycin A (MedChemExpress, HY-A0122) for 24 h and harvested for RNA isolation.

Co-inhibition of DNMT and HDAC in the five cell lines was performed as described in refs. [5,18]. Cells were treated with 500 nM/L DAC. Media containing DAC was replenished daily for 3 days. Cells were treated with 500 nM/L SB939 for 18 h and collected for RNA isolation and ChIP-seq.

For co-inhibition of DNMT and SETDB1, GP5d and OE19 cells were treated with 500 nM/L DAC. Media containing DAC was replenished daily for 3 days. Cells were treated with 500 nM/L Mitramycin A for 24 h and collected for RNA isolation.

For co-inhibition of HDAC and SETDB1, GP5d and OE19 cells were treated with 500 nM/L SB939 for 18 h. Cells were treated with 500 nM/L Mitramycin A for 24 h and collected for RNA isolation.

For p53 reintroduction experiment in GP5d p53-KO cells, GP5d p53-KO cells (clone #1) were seeded to 6-well plate (300,000 cells per well). Next day cells were transfected with p53 expressing pIRES2-EGFP-p53 WT plasmid (addgene # 49242). 24 h after transfection, cells were treated with combination of DAC (72 h) and SB939 (18 h) for co-inhibition of DNMT and HDAC or DMSO control (96 h) and collected for RNA isolation.

For DMSO control, cells were treated with DMSO (Fisher, BP231). Media containing DMSO were replenished daily for 4 days. Cells were collected for RNA isolation after 4 days.

RNeasy Mini kit (Qiagen) was used to isolate total RNA from different CMEs or DMSO treated cells. RNA-seq libraries were prepared using 500 ng of total RNA by using KAPA stranded RNA-seq kit for Illumina (Roche) as per manufacturer's instructions. All RNA-seq samples were sequenced paired-end on NovaSeq 6000 (Illumina).

## ChIP-seq analysis
The ChIP-seq reads were mapped with bowtie2 v.2.4.1 (bowtie2 --very-sensitive) to the reference human genome (hg38/GRCh38)[67]. Duplicates were removed by using Picard v.2.23.4 (MarkDuplicates -REMOVE_DUPLI-CATES false -ASSUME_SORT_ORDER coordinate) [http://broadinstitute.github.io/picard/]. Samtools v.1.7 was used to filter reads with MAPQ smaller than 20 and remove marked duplicates (samtools view -F 1024 -b -q 20)[68]. Peaks were called with MACS2 v.2.2.7.1[69]. Peaks overlapping with ENCODE blacklisted regions were removed from the peak and summit files with bedtools v.2.29.2 (bedtools subtract -A). RPKM-normalized bigwig file was prepared by using deepTools v.3.5.0 (bamCoverage --binSize 50 --normalizeUsing RPKM)[70]. Pearson correlation analysis between the biological replicates for ChIP-seq is shown in Supplementary Fig. 19a.

## ATAC-seq analysis
GP5d ATAC-seq data were acquired from ref. [5] and OE19 ATAC-seq data were acquired from ref. [38]. The ATAC-seq reads were mapped with bowtie2 v.2.4.1 (--very-sensitive) to the reference human genome (hg38/GRCh38). Reads mapped to the mitochondrial genome were removed with removeChrom.py script ([https://github.com/jsh58/harvard/blob/master/removeChrom.py](https://github.com/jsh58/harvard/blob/master/removeChrom.py)). Duplicate reads were removed by using Picard v.2.23.4 (MarkDuplicates -REMOVE_DU-PLICATES false -ASSUME_SORT_ORDER coordinate) and insert sizes were analyzed by using CollectInsertSizeMetrics. Samtools v.1.7 was used to remove marked duplicates and filter reads with MAPQ smaller than 10 (samtools view -F 1024 -b -q 10). ATAC-seq peaks were called with MACS2 v.2.2.7.1 (macs2 callpeak -f BAMPE -g hs—keep-dup all). Removal of peaks overlapping blacklisted region and preparation of a RPKM normalized bigwig was performed as in ChIP-seq data processing.

## CUT&TAG analysis
OE19 CUT&TAG data for H3K27me3, H3K4me1, RNA-Pol II and Serine-5-phosphohorylated RNA-Pol II was acquired from ref. [38]. The CUT&TAG paired-end reads were mapped with bowtie2 v.2.4.1 to the reference human genome by using following parameters: --very-sensitive --no-mixed --no-discordant -I 10. Duplicate reads were removed by using Picard v.2.23.4 (MarkDuplicates -REMOVE_DUPLICATES false - ASSUME_-SORT_ORDER coordinate) [http://broadinstitute.github.io/picard/]. Samtools v.1.7 was used to filter reads with MAPQ smaller than 20 and remove duplicates (samtools view -F 1024 -b -q 20)[68]. RPKM-normalized bigwig file was prepared by using deepTools v.3.5.0 (bamCoverage --binSize 50 --normalizeUsing RPKM)[70]. RPKM normalized bigwig files were used to plot CUT&TAG signal.

## KAS-seq analysis
OE19 KAS-seq raw data were downloaded under ENA accession "PRJEB50427" ("ERR8135308"). KAS-seq reads were aligned to the hg38 genome assembly using bowtie2 v.2.4.1 (--very-sensitive). Duplicate reads were removed by using Picard v.2.23.4 (MarkDuplicates -REMOVE_-DUPLICATES false - ASSUME_SORT_ORDER coordinate) [http://broadinstitute.github.io/picard/]. Samtools v.1.7 was used to remove marked duplicates and filter reads with MAPQ smaller than 20 (samtools view -F 1024 -b -q 20). Preparation of a RPKM normalized bigwig was performed as in ChIP-seq data processing.

## TEtranscripts and telescope RNA-seq analysis
The RNA-seq reads were mapper with STAR v.2.5.3a by using the SQuIRE pipeline v.0.9.9.92[71]. SQuIRE alignment output was used to TE subfamilies expression measurement by using TEtranscripts v.2.2.1 with following flags: –mode multi –stranded reverse. DESeq2 v.1.32.0 was used for differential expression analysis of TE subfamilies. Telescope v.1.0.3.1[72] analysis was performed on SQuIRE alignment output by using the "telescope assign" command. DESeq2 v.1.32.0 was used for differential expression analysis for individual TE loci. GREAT v.4.0.4[73] was used to assign nearby gene for TE loci.

To compare TE subfamily expression changes between cell lines (shown in Figs. 1c and 5a), TE subfamilies with strong transcriptional changes were filtered out. TE subfamilies with absolute log2FC of more than 2.5 (treatment vs DMSO control) and adjusted *p*-value less than 0.05 in at least one CMEs treatment were filtered out and expression changes in log2FC was plotted by using Pheatmap [https://www.rdocumentation.org/packages/pheatmap/versions/0.2/topics/pheatmap]. Rows and columns are clustered with hierarchical clustering. Heatmap to compare TE subfamilies expression changes for SETDB1 inhibited/KO cell lines (shown in Fig. 5a) was plotted as described in Fig. 1c by filtering out TE subfamilies with absolute log2FC of more than 1.5 (treatment vs DMSO control) and adjusted *p*-value less than 0.05.

## Nanopore data analysis

Nanopore sequencing data in GP5d cells was acquired and processed as in ref. 5. GP5d DNMTi-HDACi data was basecalled with Dorado v7.3.11 (github.com/nanoporetech/dorado) with the 400 bps super-accurate basecalling model including modified basecalling for 5mC and 5hmC. The methylBAM output files were aligned with Dorado aligner to GRCh38 with the default parameters and CpG methylation was extracted with modkit v0.3.1 (github.com/nanoporetech/modkit). The resulting bed file was loaded into R 4.2.3 with bsseq v1.34.0, using a modified read.modkit function from the development branch of bsseq (https://github.com/hansenlab/bsseq). The bsseq object was combined with the GP5d bsseq object and smoothed with bsmooth with default parameters. For comparing the CpG methylation levels, compareRegions function from ref. 60 was used.

## IR-Alu expression analysis

The genomic coordinate bed file for IR-Alus in the human genome reported in ref. 21 was shared by Dr. Parinaz Mehdipour and genomic coordinates were converted from hg19 to hg38 assembly by using liftOver tool [https://hgdownload.soe.ucsc.edu/goldenPath/hg38/liftOver/]. DESeq2 normalized Telescope expression counts were extracted for all IR-Alus and actively transcribed IR-Alu were filtered out (Total sum of RNA-seq reads for DMSO and DNMTi-HDACi ≥5) and their expression were compared for GP5d and OE19 cells with and without DNMTi-HDACi treatments.

## Alu editing index analysis

The RNA-seq reads were mapped to human reference genome with STAR v.2.7.5a with parameter recommended in ref. 42 (--outFilterMatchNminOver Lread 0.95). Duplicate reads were removed by using Picard v.2.23.4 (MarkDuplicates -REMOVE_DUPLICATES true -ASSUME_SORT_ORDER coordinate). Deduplicated bam files were sorted by using Samtools v.1.7. Alu editing index analysis was performed by using RNAeditingIndexer[42]. A to G Editing counts for exons, intron and intergenic regions were used to calculate the distribution of Alu editing sites.

## Chimeric transcript analysis

TE-chimeric transcript analysis was performed by using TEprof2 pipeline[16]. The RNA-seq reads were mapped to human reference genome with STAR v.2.7.5a and assembled with Stringtie v.1.3.3. TEprof2 v.0.1 was used to identify transcript overlapping with TEs and transcripts from GENCODE v.25. TE-chimeric transcripts were identified using TEProf2 pipeline with default parameters. TE-chimeric transcripts expressed at least two replicates in either control or treatment samples were selected for downstream analysis (Supplementary Data 6). Mean transcript expression was compared between DMSO control and DNMTi-HDACi cells for three cell lines.

## Western blotting

Cells were lysed in RIPA buffer containing 1 mM DTT (Thermo scientific, 20290) and protease inhibitor (Roche, 11873580001). Protein samples (50 μg for each sample) were denatured by using 6x SDS-Laemmli buffer (Fisher, 50-103-6570) at 95–100 °C for 5 min. Proteins were separated by SDS-PAGE using acrylamide gels (Biorad, 4569033) and transferred to PVDF membrane (Thermo scientific, 88518). Membrane was blocked in in 5% skimmed milk containing 1x TBST buffer and incubated with the following primary antibody: p53 (Santa Cruze Biotechnology, sc-126, 1:1000), Phospho-IRF7-Ser477 (St John's Laboratory, STJ196333, 1:1000), ADAR1 (Proteintech, 14330-1-AP, 1:1000), MDA5 (Proteintech, 21775-1-AP, 1:1500), SETDB1 (Proteintech, 11231-1-AP, 1:1500), H3K27ac (Abcam, ab4729, 1:1500), HDAC3 (Abcam, Ab7030, 1:1000), Histone H3ac (pan-acetyl) (Active Motif, 61638, 1:1500), Histone H3 (Proteintech, 17168-1-AP, 1:1000), DNMT1 (Proteintech, 24206-1-AP, 1:1000), IRF7 (Proteintech, 22392-1-AP, 1:1500), H3K9me3 (Diagenode, C15410193, 1:2000), and GAPDH (Santa Cruze Biotechnology, SC-47724, 1:1000). As secondary antibody, goat anti-mouse IgG (Bio-Rad, 5178-2504, 1:5000) and mouse rIgG (Bio-RAD, 5196-2504,1:5000) was used. PVDF membranes were imaged using Image Studio Lite (Odyssey CLx imager, Li-COR Biosciences).

Densitometry analysis for DNMT1 western blots was performed by using ImageJ v.1.54. Uncropped images for all western blots shown in Supplementary Fig. 20a.

## dsRNA immunostaining

Immunofluorescence was performed as previously described[21]. GP5d and OE19 cells were seeded to 8-well chamber slide (25,000 cells per well). Next day, cells were treated with DMSO for control, Mitramycin A for SETDB1 inhibition, or a combination of DAC and SB939 for co-inhibition of DNMT and HDAC. GP5d p53-KO (Clone #1) were treated with DMSO for control, DAC for DNMT inhibition, or a combination of DAC and SB939 for co-inhibition of DNMT and HDAC. Cells were washed with PBS and fixed with cold methanol for 15 min at −20 °C. Cells were washed three times with ice-cold PBS and incubated with blocking buffer (PBS with 1% BSA) for 1 h at 37 °C. Cells were incubated with anti-dsRNA primary antibody (Anti-dsRNA, clone rJ2) (Sigma, MABE1134) overnight at 4 °C. Cells were washed three times with ice-cold PBS and incubated with secondary antibody (CoraLite488-conjugated Goat AntiMouse IgG(H + L)) (Proteintech, SA00013-1) for 1 h at room temperature at 1:1000 dilution and washed three times with ice-cold PBS. Coverslips were mounted on a slide using DAPI containing mounting media (ab104139, Abcam). Microscopy was performed by using Zeiss Axio Imager at 40x with oil-immersion lens. For GP5d p53-KO cells, microscopy was performed by using Zeiss LSM800 Airyscan Microscope at 40x water-immersion lens. All captured images were analyzed with ZEISS ZEN v.3.10. The quantification of dsRNA on microscopic images was performed by using ImageJ v.1.54. GP5d cells were transfected with plasmids expressing ADAR1-targeting gRNA cloned into the Cas9-expressing plasmids (Addgene # 158115 and #158116) by using Lipofectamine 2000 (Thermo Scientific, 11668019). 24 h after transfection, cells were selected for Puromycin (2 μg per ml) for 48 h. Puromycin selected cells were used for dsRNA immunostaining and RNA isolation.

## qRT-PCR

For analyzing the expression of derepressed LTR12C nearby genes and ISGs genes by using qRT-PCR, RNA was isolated from DNMTi-HDACi, SETDB1i and DMSO treated GP5d and OE19 cells by using RNeasy Mini kit (Qiagen). cDNA synthesis was performed using the PrimeScript™ RT Master Mix (Takara, RR036A) and real-time PCR was performed using SYBR Green I Master (Roche, 04707516001) in triplicates. The primers used for each transcript are listed in Supplementary Data 9. The transcript levels of the target genes were normalized to GAPDH mRNA levels.

## RNA-seq analysis

FASTQC was used for the quality control of raw sequencing FASTQ files [http://www.bioinformatics.babraham.ac.uk/projects/fastqc/]. RNA-seq reads were aligned to the human reference genome using STAR aligner v.2.7.5c with default parameters[74] and Samtools v.1.7 was used to sort bam files. Gene counts were quantified by using HTSeq-count v.0.11.2[75]. DESeq2 v.1.32.0 was used to identify differential expressed genes for each CMEs treatment as compared to DMSO control, a threshold of |log2FC| > 1.5 and adjusted p < 0.05 was applied. Gene Set Enrichment Analysis was performed by using GSEA v.4.1.0[76].

## TCGA ATAC-seq analysis

Cancer type-specific ATAC-seq peak sets for COAD and ESCA TCGA cancer types were acquired from ref. 49. The overlap analysis with derepressed TEs was performed by using bedtools v.2.29.2.

## p53retriever analysis

All RepeatMasker LINE and LTR TE sequences were analyzed for p53REs with p53retriever[34] v.1.2 ("p53sf" with default parameters). The p53retriever output was combined with the GP5d WT, p53 KO, and OE19 untreated vs. DNMTi-HDACi DEseq2 expression analysis results from Telescope. TEs that did not have expression results from DESeq2 were discarded, and TEs with no p53REs were assigned the grade 0. In case of TE sequences with

multiple p53REs with different grades (1-5), the highest grade was assigned to the TE to avoid multiple comparisons of the same TE locus.

## Motif analyses

Motif enrichment analysis at derepressed TEs was performed using AME from MEME suite v. 5.0.2 with shuffled sequences as background (ame --control --shuffle--)[77]. JASPAR 2022 CORE non-redundant vertebrate motif annotations were used as input for the motif file. Motif clustering data were acquired from Viestra et al.[78]. The E-values were –log10 transformed by using R and for each treatment, minimal E-value for individual motifs were selected and the columns were scaled using the R scale function (center = F) and plotted.

P53 motif enrichment at derepressed TE sequences was performed by using FIMO from MEME suite v.5.0.2 by using default parameters. P53 motifs files were obtained from ref. 79. Percentage of derepressed TEs with one or more p53 motifs were compared between three cell lines in Supplementary Fig. 5h.

## Statistical analysis and plots

All statistical analyses were performed by using R v.4.1.2 and GraphPad prism v.9. Genomic annotation for derepressed TEs was performed by using ChIPseeker[80]. Boxplots were prepared by using ggplot2 v.3.3.6 from the Tidyverse suite v.1.3.1[81]. Heatmap for the motif enrichments were plotted with ComplexHeatmap v.2.8.0[82]. Heatmap in Fig. 1c and Fig. 5a was plotted by using Pheatmap [https://www.rdocumentation.org/packages/pheatmap/versions/0.2/topics/pheatmap]. Correlation analysis between replicates was performed by using multiBigwigSummary v.3.1.3 and heatmaps were plotted using plotCorrelation v.3.1.3[70]. Heatmaps and average profile plots were plotted by using deepTools[70]. Genome browser snapshots for TE-chimeric transcripts were plotted by using ggsashimi v.1.1.5[83]. All motif enrichment heatmaps and genomic annotation bar graphs were plotted by using previous deposited R scripts from ref. 5. Illustrations were created with BioRender.com.

## Statistics and reproducibility

All RNA-seq was performed in three biological replicates that were used for all analyses on RNA-seq data. The statistical tests are described for each analysis in the Methods and respective figure legends. All ChIP-seq were performed in two biological replicates. Pearson correlation analysis for ChIP-seq replicates is shown in Supplementary Fig. 19a. All qRT-PCR experiments were performed on three biological replicates. Exact $p$ values are shown in each individual figure and the source data for all relevant figures in the Supplementary Data 11.

## Reporting summary

Further information on research design is available in the Nature Portfolio Reporting Summary linked to this article.

## Data availability

Data generated in this study has been deposited in the GEO database under accession "GSE254242".

The publicly available data was accessed as follows: GP5d ChIP-seq data for H3K27ac ("GSM5454417"), H3K27me3 ("GSM5454428"), and p53 ("GSM5454412") were acquired from GEO database under accession "GSE180158". GP5d ATAC-seq ("GSE221051"), ChIP-seq for H3K4me3 ("GSM6841187"), rabbit IgG ("GSM6841190") and mouse IgG ("GSM6841189"), NaNOMe-seq ("GSM7024433"), and RNA-seq for DMSO treated ("GSM6841203", "GSM6841204", "GSM6841205") and DNMTi-HDACi treated GP5d cells ("GSM6841206", "GSM6841207", "GSM6841208") was acquired from GEO database under accession "GSE221053". GP5d H3K4me1 ("GSM1240814") was obtained with the GEO accession "GSE51234". OE19 ATAC-seq ("ERR1698333") was downloaded with accession code ERX1767841. OE19 KAS-seq ("ERR8135308") was acquired from accession code "E-MTAB-11356".

OE19 CUT&TAG data for H3K27me3 (ERR8105268), H3K4me1 (ERR8105270), RNA PolII (ERR8105275) and RNA PolIISp5 (ERR8105277) was obtained from ENA accession "E-MTAB-11356". A375 RNA-seq ("GSM5320279", "GSM5320280", "GSM5320281") and A375 SETDB1-KO RNA-seq ("GSM5320275", "GSM5320276", "GSM5320277") was acquired from GEO accession "GSE155972".

Following supplementary files provided with paper: (i) Supplementary Figs.: Supplementary Figs. and their legends as a single PDF (ii) Supplementary Data 1–10: an Excel file containing Supplementary Data Tables 1–10 as individual sheets and (iii) Supplementary Data 11: an Excel file containing source data for all graphs.

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

## Acknowledgements

We thank HiLIFE research infrastructures including the FIMM NGS Genomics laboratory at the University of Helsinki. We thank the Center for Scientific Computing (CSC), Finland, for the computational infrastructure, Professor Lauri Aaltonen's laboratory facilities for genomics work, and the Biomedicum Imaging Unit for microscopy services. We thank Norwegian Sequencing Centre, Oslo, Norway, for sequencing services. B.S. was supported by the Norwegian Centre for Molecular Biosciences and Medicine (NCMBM), University of Oslo, Norwegian Cancer Society (274630), South-Eastern Norway Health Authority (HSØ) (2025083), Research Council of Finland (320114), Finnish Cancer Foundation, Sigrid Jusélius Foundation, Jane and Aatos Erkko Foundation, and iCAN Digital Precision Cancer Medicine Flagship (320185). P.P. was supported by the Research Council of Finland (356021). V.T. was supported by doctoral program of Integrative Life Sciences, University of Helsinki. K.K. was supported by iCAN Precision Cancer Medicine Flagship. We thank Dr. Parinaz Mehdipour for sharing IR-Alu sequences for human genome. We thank Dr. Liangru Fei, and Dr. Jihan Xia for technical assistance with NGS library preparation, Inga-Lill Åberg with nanopore sequencing and Dr. Wietske van der Ent with confocal microscopy. We thank Veera Erkkilä, Dr. Subhamoy Datta, Pinja Perkkiö for assistance in laboratory experiments and Nikita Poddar for technical help.

## Author contributions

D.P. and B.S. conceived the study. B.S. supervised the study. D.P. performed experiments and genomics data analysis. V.T. helped with TEs and gene expression analysis. K.K. assisted with the Nanopore data analysis. P.P. helped with interpretation and presentation of the data. D.P. and B.S. wrote the manuscript with contributions from all authors.

## Competing interests

The authors declare no competing interests.
