## [Transparent Peer Review file · Communications Biology]

Cancer cell type-specific derepression of transposable elements by inhibition of chromatin modifier enzymes

Corresponding Author: Dr Biswajyoti Sahu

This manuscript has been previously reviewed at another journal. This document only contains information relating to versions considered at Communications Biology.

Version 0:

Reviewer comments:

Reviewer #4

(Remarks to the Author)

In this manuscript, Patel and colleagues propose a mechanism for the derepression of transposable elements (TEs) by epigenetic therapies in a cancer cell-type-specific manner. Using GP5d colorectal, OE19 oesophageal, and LNCap-1F5 prostate cancer cell lines, they demonstrate the reactivation of distinct TEs in response to different epigenetic therapies. While the study presents interesting findings, it lacks a clear focus and does not sufficiently explore the underlying mechanisms driving the observed phenotypes. This makes it challenging to assess the novelty of the work in comparison to previous studies that have already reported the effects of epigenetic therapies or p53 mutations on TE regulation. As a result, the study does not appear to represent a significant advance or provide enough new insights beyond what has already been published in this field.

I have a few major concerns regarding the data presented.

Major Points:

1. Although the authors included a cell viability assay and Western blot analysis in Supplementary Figure 2A and 2B to demonstrate the on-target effects of the epigenetic therapies used, it is surprising that none of the inhibitors in either of the two cancer cell lines reduce cell viability by more than 75%, even at a 1 mM dose.

Additionally, the Western blot results showing DNMT1 protein reduction upon DAC treatment are not entirely convincing, despite the inclusion of the densitometry plot.

To strengthen the analysis of histone modifications, it would be important to include total H3 as a loading control. And it is not convincing based on these two assays how the 500nM dose was selected for downstream assays.

Given these observations, have the authors examined the mutation status of DNMT1, SETDB1, and key HDAC enzymes in these two cell lines? This could provide further insight into the responsiveness of these cells to epigenetic therapies and the observed differences on basal protein levels of them.

2. In supplementary Figure 2C, it does not appear to be a significant additive induction of TEs in the DNMTi-HDACi combination treatment in either WT or p53KO conditions.

3. In Supplementary figure 10, GP5d cells do not show detectable protein levels of ADAR1 for either the p110 or p150 isoform. Additionally, there is no Western blot data demonstrating ADAR1 depletion in these cells, which already express very low levels of ADAR1. Furthermore, the manuscript does not provide an explanation or potential mechanism for the observed phenotype regarding the effect of the tested epigenetic therapies on ADAR1 expression. Clarifying this aspect would strengthen the study's conclusion

4. The data presented in Supplementary Figure 10 is not entirely convincing, as SETDB1i treatment in OE19 appears to reduce p150 expression. Additionally, the manuscript does not provide a clear underlying mechanism for the reduction of

ADAR1 expression observed with the HDACi-DNMTi combination treatment. Further clarification on this point would strengthen the study's conclusions

Minor points:

The word Invested in line 428 should be change to inverted.

Version 1:

Reviewer comments:

Reviewer #5

(Remarks to the Author)

According to what was requested from me for this paper, I am providing my opinion regarding the comments raised by reviewer 4 for this paper.

I shared most of his/her initial concerns. I feel that the authors have to some extent addressed all the points that were initially raised.

Although the manuscript has an extensive reference list, I feel that there are several important references that should be added / discussed:

- A discussion of the recent Immunity study by the group of Ben Greenbaum (Sun et al, Immunity, 2024) on differential regulation of TE immunogenicity depending on the p53 status.

- The authors discuss occurrence of chimeric transcripts upon treatment with Setdb1 inhibitor, but they don't discuss a recent study that formally establish that Setdb1 inactivation / expression levels is associated to chimeric TE-exon transcripts (Burbage et al, Science Immunology, 2023).

Response to reviewers

Reviewer #4 (Remarks to the Author):

In this manuscript, Patel and colleagues propose a mechanism for the derepression of transposable elements (TEs) by epigenetic therapies in a cancer cell-type-specific manner. Using GP5d colorectal, OE19 oesophageal, and LNCap-1F5 prostate cancer cell lines, they demonstrate the reactivation of distinct TEs in response to different epigenetic therapies. While the study presents interesting findings, it lacks a clear focus and does not sufficiently explore the underlying mechanisms driving the observed phenotypes. This makes it challenging to assess the novelty of the work in comparison to previous studies that have already reported the effects of epigenetic therapies or p53 mutations on TE regulation. As a result, the study does not appear to represent a significant advance or provide enough new insights beyond what has already been published in this field.

We thank the reviewer for reviewing our manuscript and interest in our results. We have addressed reviewer's concerns in the revised manuscript and clarified the manuscript text to better highlight the novelty of our results.

Our findings are based on five distinct cell lines representing three different tissues of endodermal origin with different p53 status, enabling a systematic analysis of TE regulation by CMEi treatments. Even though derepression of TEs by epigenetic therapies or p53 mutations has been reported earlier in individual cell lines, a comparative analysis of four distinct CME inhibition treatments on TE expression in different cell types with and without functional p53 has not been addressed earlier. We have now modified the manuscript to highlight this (page 25 line 618-626). This is important due to distinct cancer cell type-specific responses and p53 mutations in most cancer types. Our novel findings compared to previous reports include: i) A strong effect of p53 in potentiating the effect of CMEi treatments on TE activation and TE-chimeric transcript expression (page 10, lines 222-230, page 21, lines 523-527, and page 27, lines 666-669), ii) Differential regulation of ADAR1 by different CMEi treatments, resulting in either decreased Alu RNA editing in response to DNMTi-HDACi, or increased Alu editing in the case of SETDB1i (page 18-19, lines 452-464). Differential regulation of ADAR1 by different CMEi treatments and their effect on Alu editing index was not reported earlier except for DNMT inhibition. and iii) TEs regulation by novel treatment combinations by co-inhibition of DNMT and SETDB1, and HDAC and SETDB1 (page 8, lines 179-183, and page 19, lines 468-470). Collectively, we feel that our manuscript represents a systematic comparison of the different CMEs on TE regulation in p53 wild-type and mutant cancers.

Major Points:

1. Although the authors included a cell viability assay and Western blot analysis in Supplementary Figure 2A and 2B to demonstrate the on-target effects of the epigenetic therapies used, it is surprising that none of the inhibitors in either of the two cancer cell lines reduce cell viability by more than 75%, even at a 1 mM dose.

We thank the reviewer for this comment. We have now clarified the manuscript text referring to **Supplementary Figure 1** to emphasize that the treatments did not induce considerable cytotoxicity at the concentration used in this study (page 5, line 104-105). However, the larger doses resulted in reduced cell viability comparable to previous reports as outlined below. We have revised the legend for **Supplementary Figure 1** to clarify this.

(i) DAC treatment at 10 μ M dose was reported to result in less than 25% decrease in the viability of six colon cancer cell lines (PMID: 35326680), which is comparable to 20% and 19% reduction that we observed at the same dose in GP5d and OE19 cells, respectively. At 1 mM dose, DAC reduced cell viability by 77% and 76% in GP5d and OE19 cells, respectively. (ii) SB939 treatment at 0.1 mM dose was reported to result in 35% to 75% reduction in the viability of four breast cancer cell lines (PMID: 32109485), comparable to 40% reduction that we observed in GP5d and OE19 cells at 0.1 mM, whereas 1 mM dose induced greater than 50% reduction in both our cell lines. (iii) Mitramycin A treatment at 1 mM dose was reported to reduce cell viability by 50% and 25% in two melanoma lines (PMID: 32637583), whereas in GP5d and OE19 cells we observed a stronger effect on cell viability with greater than 50% reduction.

Additionally, the Western blot results showing DNMT1 protein reduction upon DAC treatment are not entirely convincing, despite the inclusion of the densitometry plot.

We thank the reviewer for pointing this out and apologise for the poor quality of the blot. We have now repeated the DNMT1 Western blot using a higher amount of total protein. Revised **Supplementary Fig. 2a** now shows the reduction in DNMT1 protein level upon DAC treatment both in GP5d and OE19 cells. We have removed the densitometry plot from the figure.

The likely reason for moderate reduction in the DNMT1 level upon DAC treatment is its mechanism of action. DAC is a nucleotide analogue prodrug that induces replication-dependent DNA hypomethylation by trapping DNMT1 in the progressing replication fork (PMID: 18398832), but it does not reduce the *de novo* synthesis of DNMT1 (PMID: 18398832). However, we have validated the on-target effect of DAC-treatment by directly measuring DNA methylation using long-read nanopore sequencing, and the DAC-induced hypomethylation is shown in **Fig. 4f** and **Supplementary Fig. 8**. We have now clarified these aspects in the revised manuscript (page 26, lines 639-643).

To strengthen the analysis of histone modifications, it would be important to include total H3 as a loading control. And it is not convincing based on these two assays how the 500nM dose was selected for downstream assays.

We thank the reviewer for this important suggestion. We have now included Histone H3 as a loading control for histone modification Western blots in **Supplementary Figure 2a**. We have modified manuscript text on page 35, line 938 and figure legend for **Supplementary Figure 2a** to address this change. Regarding the 500 nM dose chosen for the experiments, we have now revised the manuscript text to clarify that the doses of each drug were based on earlier studies (PMID: 37658059, 37872136, 28604729, 32109485, 32637583) (page 4, lines 98-101).

Given these observations, have the authors examined the mutation status of DNMT1, SETDB1, and key HDAC enzymes in these two cell lines? This could provide further insight into the responsiveness of these cells to epigenetic therapies and the observed differences on basal protein levels of them.

We thank reviewer for this important suggestion. We have now included mutation details for the CME in new **Supplementary table 7** and discussed this in the revised manuscript on page 26 and lines 631-637. For GP5d and LNCaP cells, (i) one allele of HDAC4 had frameshift mutation causing loss of function and (ii) missense mutation was reported for

DNMT3A, DNMT3B, HDAC1, HDAC3 and SETDB1. There was no mutation reported in DNMT, SETDB1 and HDAC enzymes in OE19 cells.

2. In supplementary Figure 2C, it does not appear to be a significant additive induction of TEs in the DNMTi-HDACi combination treatment in either WT or p53KO conditions.

We thank reviewer for this comment. To clarify the additive induction of TEs by DNMTi-HDACi shown in **Supplementary Fig. 2b**, we have now listed the numbers of up- and down-regulated TE subfamilies by CMEi treatments in new **Supplementary Data 1**. We have also clarified this point in manuscript text on **page 5, lines 122-126**, indicating that the synergistic effect of DNMTi-HDACi is based on 1.5- to 3.2-fold higher number of upregulated TE subfamilies compared to the sum of subfamilies induced by DNMTi and HDACi individually.

3. In Supplementary figure 10, GP5d cells do not show detectable protein levels of ADAR1 for either the p110 or p150 isoform. Additionally, there is no Western blot data demonstrating ADAR1 depletion in these cells, which already express very low levels of ADAR1. Furthermore, the manuscript does not provide an explanation or potential mechanism for the observed phenotype regarding the effect of the tested epigenetic therapies on ADAR1 expression. Clarifying this aspect would strengthen the study's conclusion

We thank the reviewer for this important point. We agree that the interpretation of ADAR1 Western blot was difficult in GP5d cells due to lower baseline expression levels. We have now repeated the ADAR1 Western blot by using a higher amount of total protein and show ADAR1 protein reduction by DNMTi-HDACi treatment in the revised **Supplementary Fig. 10d**. The lower baseline expression could stem from the nonsense mutation in one of the ADAR1 alleles in GP5d cells, which is now clarified on **page 18-19, line 453-458**.

We agree that the mechanism of ADAR1 downregulation by DNMTi-HDACi is an important question. Interestingly, earlier reports have implicated BTRC E3 Ubiquitin ligase in degradation of p110 ADAR1 isoform in response to IFN signalling (PMID: 27729454), and we have noted this as a potential mechanism for ADAR1 downregulation in the Discussion of the revised manuscript (**page 28, line 706-710**). Indeed, BTRC was upregulated in DNMTi-HDACi treated GP5d and OE19 cells (**Editorial Fig. 1**). Furthermore, ADAR1 was reported to interact with BTRC, HDAC5 and HDAC9 (<https://thebiogrid.org/106617/summary/homo-sapiens/adar.html>). Thus, DNMTi-HDACi induced BTRC expression might contribute to ADAR1 downregulation. However, to understand mechanistic details of complex interplay between ADAR1 and BTRC in response to distinct CMEi treatments needs additional investigation in the future, as we have noted in the Discussion (**page 28, line 706-711**).

Editorial Fig. 1: Comparison of normalized RNA-seq read counts for β transducin repeat-containing protein (*BTRC*) in GP5d, and OE19 cells treated with CME inhibitors (Unpaired two-sided t-test).

4. The data presented in Supplementary Figure 10 is not entirely convincing, as SETDB1i treatment in OE19 appears to reduce p150 expression. Additionally, the manuscript does not provide a clear underlying mechanism for the reduction of ADAR1 expression observed with the HDACi-DNMTi combination treatment. Further clarification on this point would strengthen the study's conclusions

We thank the reviewer for this interesting comment. Indeed, SETDB1i treatment reduced p150 isoform expression in OE19 cells but increased p110 expression in GP5d. Both isoforms have a higher baseline expression level in OE19 cells, and thus the effect of SETDB1i on p110 upregulation was weaker. We have now clarified this on page 19, line 458-461. Contribution of each isoform to the editing landscape is not yet fully understood, but interestingly, it has been reported that more than half of the A-to-I edit sites are selectively edited by p150, and the other half are edited by either p150 or p110 in HEK293T cells (PMID: 33723056). This along with our observations about the effect of SETDB1i on ADAR1 expression highlight the complexity of these pathways, warranting future mechanistic studies in different cell lines and addressing the common and isoform-specific effects of RNA editing. We have now revised the Discussion to highlight this point (page 28, line 701-705).

As mentioned to our reply to major point 3, the potential mechanism for reduced ADAR1 expression warrants additional investigation in the future, as we have noted in the Discussion (page 28, line 706-711).

Minor points:

The word Invested in line 428 should be change to inverted.

We thank the reviewer for pointing out this typo and have corrected it.

We have highlighted changes in manuscript text file.

Response to reviewers

Reviewer #5 (Remarks to the Author):

According to what was requested from me for this paper, I am providing my opinion regarding the comments raised by reviewer 4 for this paper.

I shared most of his/her initial concerns. I feel that the authors have to some extent addressed all the points that were initially raised.

We thank the reviewer for their positive feedback on our revisions to address the comments by Reviewer 4.

Although the manuscript has an extensive reference list, I feel that there are several important references that should be added / discussed:

- A discussion of the recent Immunity study by the group of Ben Greenbaum (Sun et al, Immunity, 2024) on differential regulation of TE immunogenicity depending on the p53 status.

- The authors discuss occurrence of chimeric transcripts upon treatment with Setdb1 inhibitor, but they don't discuss a recent study that formally establish that Setdb1 inactivation / expression levels is associated to chimeric TE-exon transcripts (Burbage et al, Science Immunology, 2023).

We thank the reviewer for this valuable comment. We have now included these references on page 8, lines 219-220, page 11, lines 346-347 and page 16, lines 501-502.